# IMPROVING NEURAL NETWORK ACCURACY AND CALIBRATION UNDER DISTRIBUTIONAL SHIFT WITH PRIOR AUGMENTED DATA

## ABSTRACT

Neural networks have proven successful at learning from complex data distributions by acting as universal function approximators. However, neural networks are often overconfident in their predictions, which leads to inaccurate and miscalibrated probabilistic predictions. The problem of overconfidence becomes especially apparent in cases where the test-time data distribution differs from that which was seen during training. We propose a solution to this problem by seeking out regions in arbitrary feature space where the model is unjustifiably overconfident, and conditionally raising the entropy of those predictions towards that of the Bayesian prior on the distribution of the labels. Our method results in a better calibrated network and is agnostic to the underlying model structure, so it can be applied to any neural network which produces a probability density as an output. We demonstrate the effectiveness of our method and validate its performance on both classification and regression problems by applying it to the training of recent state-of-the-art neural network models.

## 1 INTRODUCTION

While deep neural networks have achieved success on many diverse tasks due to their ability to learn highly expressive task-specific representations, they are known to be overconfident when presented with unseen inputs from unknown data distributions. Probabilistic models should be accurate in terms of both *accuracy* and *calibration*. Accuracy measures how often the model's predictions agree with the labels in the dataset. Calibration measures test the accuracy of the uncertainty around a probabilistic output. For example, an event predicted with 10% probability should be the empirical outcome 10% of the time. The probability around rare but important outlier events needs to be trustworthy for mission critical tasks such as autonomous driving.

Bayesian neural networks (BNN) and ensembling methods are popular ways to achieve a predictive distribution for both classification and regression models. Since Gal & Ghahramani (2015) showed that Monte Carlo Dropout acts as a Bayesian approximation, there have been numerous advances in modeling predictive uncertainty with BNNs. As laid out by Kendall & Gal (2017), models need to account for sources of both *aleatoric* and *epistemic* uncertainty. Epistemic uncertainty arises from uncertainty in knowledge or beliefs in a system. For parametric models such as BNNs, this presents as uncertainty in the parameters which are trained to encode knowledge about a data distribution. *Aleatoric* uncertainty arises from irreducible noise in the data. Correctly modeling both forms of uncertainty is essential in order to form accurate and calibrated predictions.

Accuracy and calibration are negatively impacted when the data seen during deployment varies substantially from that seen during training. It has been shown that when test data has undergone a significant distributional shift from the training data, one can witness performance degradation across all models (Snoek et al., 2019). A recurring result from Snoek et al. (2019) is that Deep Ensembles (Lakshminarayanan et al., 2017) show superior performance on shifted test data. Previous work has also shown that BNNs fail to accurately model epistemic uncertainty, as regions with sparse amounts of training data often lead to confident predictions even when evidence to justify such confidence is lacking (Sun et al., 2019). Bayesian non-parametric models such as Gaussian

processes (GP) also model epistemic uncertainties, but suffer from limited expressiveness and the need to specify a kernel *a priori* which may not be feasible for distributions with unknown structure.

In this work, we propose a new method for achieving accurate and calibrated models by providing generated samples from a variational distribution which augments the natural data to seek out areas of feature space for which the model exhibits unjustifiably low amounts of uncertainty. For those regions of features, the model is encouraged to predict uncertainty closer to that of the Bayesian prior belief. Our method can be applied to any existing neural network model during training, in any arbitrary feature space, and results in improved accuracy and calibration on shifted test data.

Our contributions in this work are as follows:

- We propose a new method of data augmentation, which we dub **Prior Augmented Data (PAD)** that seeks to generate samples in areas where the model has an unjustifiably low level of epistemic uncertainty.

- We introduce a method for creating OOD data for **regression problems**, which to our knowledge has not been proposed before.

- We experimentally validate our method on shifted data distributions for both **regression and classification** tasks, on which it significantly improves both the accuracy and the calibration of a number of state-of-the-art Bayesian neural network models.

## 2 BACKGROUND

For regression tasks, we denote a set of features $\mathbf{x} \in \mathbb{R}^d$ and labels $y \in \mathbb{R}$ which make up a dataset of i.i.d. samples $\mathcal{D} = \{(\mathbf{x}_n, y_n)\}_{n=1}^N$, with $\mathbf{X} := \{\mathbf{x}_n\}_{n=1}^N$. Let $f_\theta$ be a neural network which is parameterized by weights $\theta$. Let the output of $f_\theta(\mathbf{x})$ be a probability density $p_\theta(y|\mathbf{x})$ which is either in the form of a Gaussian density $\mathcal{N}(\mu, \sigma)$ for single task regression or a categorical distribution in the case of multi-class classification. Let $g_\phi$ be a generative model which generates pseudo inputs for $f_\theta$. We assume that both $f_\theta$ and $g_\phi$ are iteratively trained via mini-batch stochastic gradient descent with updates to generic model parameters $\tau$ given by the update rule $\tau_{t+1} = \tau_t - \nabla_{\tau_t}\mathcal{L}$, with $\mathcal{L}$ representing a loss function which is differentiable w.r.t generic parameters $\tau$. We refer to an out-of-distribution (OOD) or distributionally shifted dataset $\tilde{\mathcal{D}}$ as one which is drawn from a different region of the distribution than the training dataset $\mathcal{D}$. This distributional shift can occur naturally for multiple reasons including a temporal modal shift, or an imbalance in training data which may come about when gathering data is more difficult or costly in particular regions of $\mathcal{D}$.

### 2.1 BAYESIAN NEURAL NETWORKS

BNNs are neural networks with a distribution over the weights that can model uncertainty in the weight space. In practice, this is often done by introducing a variational distribution and then minimizing the Kullback-Leibler divergence (Kullback, 1997) between the variational distribution and the true weight posterior. For a further discussion of this topic, we refer the reader to existing works (Kingma & Welling, 2013; Blundell et al., 2015; Gal & Ghahramani, 2015). During inference, BNNs make predictions by approximating the following integral with Monte Carlo samples from the variational distribution $q(\theta|\mathcal{D})$.

$$p(y|\mathbf{x}, \mathcal{D}) = \int p_\theta(y|\mathbf{x})p(\theta|\mathcal{D})d\theta \approx \frac{1}{S}\sum_{s=1}^S p_{\theta_s}(y|\mathbf{x})q(\theta_s|\mathcal{D}), \quad \theta_1, \ldots, \theta_S \overset{\text{i.i.d.}}{\sim} q(\theta|\mathcal{D}). \quad (1)$$

### 2.2 MISPLACED CONFIDENCE

A problem arises when neural networks do not accurately model the true posterior over the weights $p(\theta|\mathcal{D})$ given in (2). Our conjecture is that a major factor contributing to generalization error in both likelihood and calibration is a failure to revert to the prior $p(\theta)$ for regions of the input space with insufficient evidence to warrant low entropy predictions. Bayesian non-parametric models such as Gaussian processes (GP) solve this through utilizing a kernel which makes pairwise comparisons between all datapoints. GP's come with the drawback of having to specify a kernel *a priori* and are

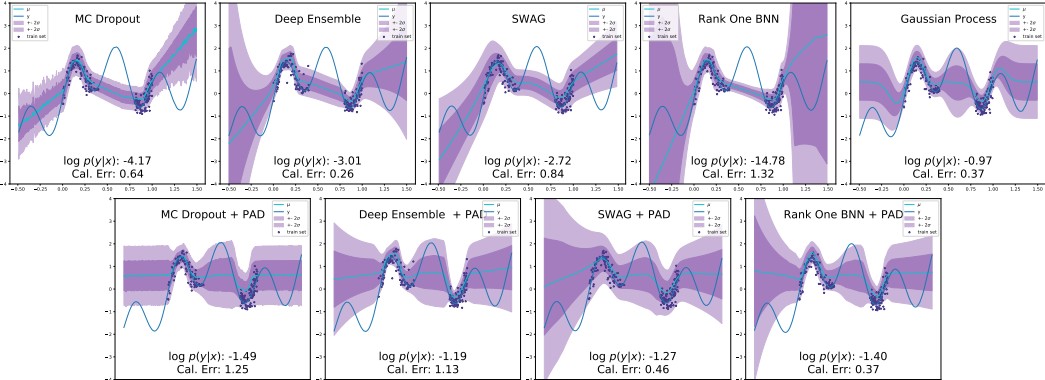

Figure 1: Performance of different models on OOD test data. The **top row** contains baseline models, the **bottom row** contains baseline + PAD. It can be seen that PAD effectively increases the epistemic uncertainty in regions with sparse training data and exhibits a more predictable reversion to the prior, much like a GP (top right). Baseline BNN models in the top row show unpredictable behavior in OOD regions of input space.

generally outperformed by deep neural networks (DNNs) which are known to be more expressive than a GP with a common RBF kernel.

$$p(\theta|\mathcal{D}) = \frac{p(\mathcal{D}|\theta)p(\theta)}{\int p(\mathcal{D}|\theta')p(\theta')d\theta'}. \tag{2}$$

The problem of misplaced confidence of neural network was first studied by Guo et al. (2017) which showed that modern DNNs have poor correlation between their confidence and actual accuracy, and are often overconfident. Snoek et al. (2019) came to the general conclusion that Deep Ensembles (Lakshminarayanan et al., 2017) tend to be the best calibrated model on OOD test data. Since then, there have been a number of newly proposed BNN models such as SWAG (Snoek et al., 2019) Multi-SWAG (Wilson & Izmailov, 2020), and Rank One Bayesian Neural Networks (R1BNN) (Dusenberry et al., 2020), each of which utilize different strategies for modeling $p(\theta|D)$.

To illustrate the problem of failing to revert to the prior in underspecified regions of input space, we have provided a toy example in (figure 1). The true function is given by $x + \epsilon + \sin(4(x + \epsilon)) + \sin(13(x + \epsilon))$, where $\epsilon \sim \mathcal{N}(0, 0.03)$. We then sample 100 points in the range of $[0, 0.4]$ and 100 points in the range of $[0.8, 1.0]$, which leaves a gap in the training data. One can observe in (Figure 1) how different neural network models tend to make confident predictions in regions where they have not observed any data, and exhibit unpredictable behavior around the outer boundaries of the dataset. Our method encourages a reversion to the prior in uncertain regions, similar to the behavior of a GP. In addition to covering the gap between the datapoints, our model exhibits more predictable behavior around the outer boundaries of the data when compared to other BNN models.

## 3 METHOD

To encourage a reversion towards the prior in uncertain regions, we learn to generate pseudo OOD data which leads to a better calibrated model by raising the entropy of the OOD predictions. An important design choice that we make is for our OOD generator network $g_\phi(\cdot)$ to take a *dataset* **X** as input to produce distributions of OOD data. The goal of the OOD generator is to fill the "gaps" between the training data, using all available current knowlege — the training data itself.

Once trained, the model $f_\theta$ should predict more uncertainty for data generated from $g_\phi$. Raising the entropy of uninformative noisy psuedo inputs may be a solution, but also could be too distant from the natural data, and therefore provide no useful information for $f_\theta$ to learn from. Ideally, we want realistic OOD data that are not too distant from the training data and still predicted with more uncertainty by $f_\theta$. To achieve this, we employ an adversarial training procedure similar to generative adversarial nets (Goodfellow et al., 2014a) — we train $g_\phi$ to find where $f_\theta$ may be overconfident,

and at the same time train $f_\theta$ to defend against this by predicting higher levels of uncertainty in those regions. In the next section, we explain the objectives of $g_\phi$.

## 3.1 THE OUT-OF-DISTRIBUTION SAMPLE GENERATOR NETWORK $g_\phi$

$g_\phi$ takes a dataset $\mathbf{X} = \{\mathbf{x}_i\}_{i=1}^n$ and produces a distribution of an equally sized pseudo dataset $\tilde{\mathbf{X}} = \{\tilde{\mathbf{x}}_n\}_{n=1}^N$. First, each $\mathbf{x}_i$ is encoded via a feedforward network $g_{\text{enc}}(\cdot)$ to construct a set of representations $\mathbf{Z} = \{\mathbf{z}_n = g_{\text{enc}}(\mathbf{x}_n)\}_{n=1}^N$. Then, we pick a subset size $K \in [1, \lfloor N/2 \rfloor]$ for each mini-batch, and for each $n = 1, \dots, N$,

$$g_\phi(\mathbf{X})_n = g_{\text{dec}}\left(\frac{1}{K} \sum_{m \in \text{nn}_K(n)} \mathbf{z}_m\right),$$ (3)

where $\text{nn}_K(m)$ denotes the set of $K$-nearest neighbors of $\mathbf{z}_n$ and $g_{\text{dec}}$ is another feedforward network. The distribution for the generated data $\tilde{\mathbf{X}}$ is then defined as

$$q_\phi(\tilde{\mathbf{X}}|\mathbf{X}) = \prod_{n=1}^N q_\phi(\tilde{\mathbf{x}}_n|g_\phi(\mathbf{X})_n).$$ (4)

As stated previously, our goal is to generate psuedo-OOD data, but we cannot be sure where such data will arise from. The only thing we can be certain of is that 1) the training data exists, and 2) there exists some distribution which is OOD to the training data. Therefore, our only option is to learn directly from representations of training data in order to find likely regions of OOD data.

## 3.2 TRAINING OBJECTIVES

**Training $g_\phi$**  Given a batch of data $\mathbf{X}_B = \{\mathbf{x}_n\}_{n=1}^B$, we first construct the OOD distribution $g_\phi(\tilde{\mathbf{X}}_B|\mathbf{X}_B)$ via $g_\phi$. The training loss for $\mathbf{X}_B$ is defined as

$$\ell_\phi(\mathbf{X}_B) = \frac{1}{B} \sum_{n=1}^N \Big( \underbrace{\mathbb{E}_{q_\phi(\tilde{\mathbf{x}}_n)}\big[\mathcal{H}[p_\theta(y|\tilde{\mathbf{x}}_n)]\big]}_{\text{A}} - \underbrace{\mathcal{H}[q_\phi(\tilde{\mathbf{x}}_n)]}_{\text{B}} \Big) + \underbrace{\frac{1}{BK} \sum_{n=1}^B \sum_{m \in \text{nn}_K(n)} \mathbb{E}_{q_\phi(\tilde{\mathbf{x}}_n)}[\|\tilde{\mathbf{x}}_n - \mathbf{x}_m\|^2]}_{\text{C}},$$

(5)

where $q_\phi(\tilde{\mathbf{x}}_n) := q_\phi(\tilde{\mathbf{x}}_n|g_\phi(\mathbf{X}_B)_n)$ and $\mathcal{H}[p] := -\int p(\mathbf{x})\log p(\mathbf{x})d\mathbf{x}$ is the entropy, and $\text{nn}_K(m)$ in C is the set of $K$-nearest neighbors of $\tilde{\mathbf{x}}_n$ among $\mathbf{X}_B$. The A term trains $g_\phi$ to fool $f_\theta$ by seeking regions where $f_\theta(\tilde{\mathbf{x}}_n)$ makes low entropy predictions, i.e., where it may be overconfident on OOD data. The B term encourages diversity of the generated samples by maximizing the entropy of $q_\phi(\tilde{\mathbf{x}}_n)$. Without B, the generated samples would be prone to mode collapse and become homogeneous and uninformative. The final term, C, minimizes the average pairwise distance between the real data and the generated data it is conditioned on, where $\mathbf{x}_m \in \tilde{\mathbf{X}}_B$. As the C term minimizes this distance, the generated data is likely to exist in the "gap" regions of the natural training data. Combined with the training loss for $f_\theta$ which we will describe next, we aim to train $g_\phi$ to produce $\tilde{\mathbf{x}}$ that are not too distant from the training data but still differ enough to warrant an increase in uncertainty. We provide an ablation study to see the effects of the terms A, B, and C, in tables 4 and 5.

Given the training data $\mathcal{D}$, we optimize the expected loss $\mathcal{L}_\phi := \mathbb{E}_{\mathbf{X}_B}[\ell_\phi(\mathbf{X}_B)]$ over subsets of size $B$ obtained from $\mathbf{X}$. In practice, at each step, we sample a single mini-batch $\mathbf{X}_B$ to compute the gradient. Also, we choose $q_\phi(\tilde{\mathbf{x}}_n)$ to be a reparameterizable distribution, and approximate the expectation over $\tilde{\mathbf{x}}_n$ via a single sample $\tilde{\mathbf{x}}_n \sim q_\phi(\tilde{\mathbf{x}}_n)$.

For regression tasks we found the constraint given by C in 5 was too strong of a constraint given that the generation happens directly in the input space and the datasets are generally of low dimensionality. For regression, we therefore only started to penalize the C term distance when it exceeded a certain boundary threshold. We used $||1^d||$, where $d$ is the dimensionality of the inputs.

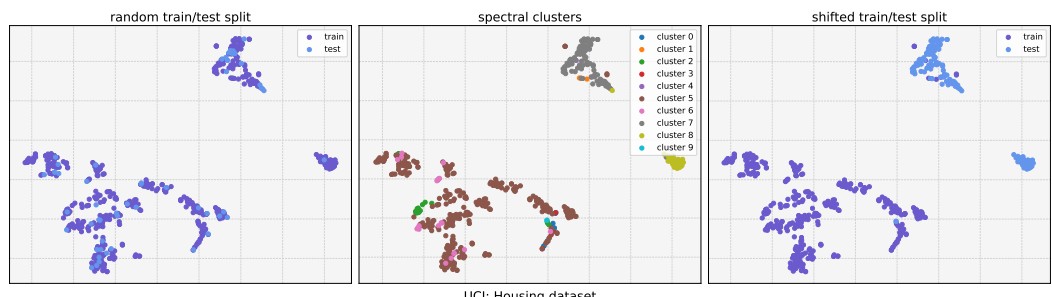

UCI: Housing dataset

Figure 2: **Left**: A random train/test split. The test data is randomly chosen from the whole distribution. **Middle**: Clusters resulting from spectral clustering. **Right**: Shifted data distributions chosen from clusters as a train/test split for our experiments on OOD data.

**Training $f_\theta$**    For $f_\theta$, we minimize the expected loss,

$$\mathcal{L}_\theta = \mathbb{E}_{(\mathbf{x},y)}[-\log p_\theta(y|\mathbf{x})] + \mathbb{E}_{\mathbf{X}_B}[r_\theta(\mathbf{X}_B)], \tag{6}$$

where the first term is the negative log-likelihood of the natural data, and the second term is a regularizer which encourages more uncertainty on OOD data,

$$r_\theta(\mathbf{X}_B) := \frac{1}{B} \sum_{n=1}^B \mathbb{E}_{q_\phi(\tilde{\mathbf{x}}_n)} \left[ \left( 1 - \exp\left( -\frac{\min_m \|\tilde{\mathbf{x}}_n - \mathbf{x}_m\|^2}{2\ell^2} \right) \right) \mathrm{KL}[p_\theta(y|\tilde{\mathbf{x}}_n)\|p(y)] \right], \tag{7}$$

where $\ell > 0$ is a scaling parameter. The role of the regularizer is to encourage predictions on the OOD data to be closer to the prior $p(y)$. Note that the KL term decays towards zero as $\mathbf{x}$ becomes closer to $\tilde{\mathbf{x}}$. This allows $f_\theta$ the freedom to make confident predictions in regions where real data exists. We draw a mini-batch $\mathbf{X}_B$ from $\mathbf{X}$, approximating $r_\theta(\mathbf{X}_B)$ with a single sample $\tilde{\mathbf{X}}_B \sim q_\phi(\tilde{\mathbf{X}}_B|\mathbf{X}_B)$. The loss is then approximated by (8),

$$\mathcal{L}_\theta \approx -\frac{1}{B} \sum_{n=1}^B \log p_\theta(y_n|\mathbf{x}_n) + r_\theta(\mathbf{X}_B) \tag{8}$$

In terms of added complexity, PAD optimizes one additional set of parameters $\phi$ and requires an alternating pattern of training akin to that of GAN's (Goodfellow et al., 2014a). In our experiments, we utilize a shallow network with a single hidden layers for $\phi$. For further information about how we implement the $\mathbb{KL}$ term in 6 we we refer the reader to section 8 for details and algorithms.

## 4    EXPERIMENTS

### 4.1    EXPERIMENTAL SETUP AND DATASETS

For regression, we use UCI datasets (Dua & Graff, 2017) following Hernández-Lobato & Adams (2015). The base network is a multi-layer perceptron with two hidden layers of 50 units with ReLU activations. The generator uses a single hidden layer of 50 units for both the encoder and decoder and uses a permutation invariant pooling layer consisting of $[mean(x), max(x)]$. The encoding of $\mathbf{X}$ and generation of $\tilde{\mathbf{X}}$ are done directly in the input space. We train the models for a total of 50 epochs on 10 random splits of train/test data. In order to create an shifted test set, we first run a spectral clustering algorithm to get 10 clusters on each dataset. We then randomly choose test clusters until we have a test set which is $\geq 20\%$ of the total dataset size, then use the remaining clusters for training. We repeat this process 10 times for each dataset. A TSNE visualization of a dataset created this way is given in (figure 2). We do this to ensure there is a significant shift between training and testing data. We report both negative log likelihood (NLL) and calibration error as proposed by Kuleshov et al. (2018) measured with 100 bins on the cumulative distribution function (CDF) of the density $p_\theta(y_i|\mathbf{x}_i)$. We show calibration plots in (figure 3), plotting the expected outcome frequency versus the empirical frequency. For baseline models, we tune hyperparameters with a randomly chosen 80/20 training/validation split. For PAD models, we do k-fold cross validation with $K = 2$, selecting half of the previously made clusters for each fold. For all models and datasets, we tune the hyperparameters for each individual train/test split.

Table 1: Negative log likelihood on UCI regression datasets. GP's and FVBNN's are included for reference. **bold** entries contain the best result for the base model. underlined entries are those with a large difference of $\geq 1$ (log scale) between methods.

| Model | Housing | Concrete | Energy | Kin8nm | Naval | Power | Wine | Yacht |
|---|---|---|---|---|---|---|---|---|
| GP | 3.81±0.23 | 4.44±0.08 | 6.74±7.44 | -0.50±0.04 | -3.55±2.07 | 4.14±0.36 | 1.22±0.05 | 4.31±0.21 |
| FVBNN | 4.62±1.08 | 4.80±0.50 | 3.16±1.02 | 0.01±0.26 | -3.19±0.40 | 3.18±0.10 | 1.30±0.00 | 3.26±0.00 |
| DUN | 4.86±1.64 | 4.78±0.73 | 4.41±1.22 | 0.85±1.49 | -1.16±0.74 | 4.34±1.22 | 3.61±7.06 | 6.37±3.10 |
| DE | 6.83±2.92 | 8.34±4.66 | 10.73±23.25 | **-1.03±0.23** | -0.42±1.92 | 3.12±0.18 | 2.18±0.67 | **2.39±0.64** |
| DE PAD | **3.61±0.22** | **5.13±0.98** | **3.24±0.62** | -0.38±0.10 | **-2.81±0.36** | 3.12±0.08 | **1.24±0.08** | 3.78±0.07 |
| R1BNN | 4.18±0.70 | 5.17±1.27 | 9.22±19.74 | **-0.86±0.19** | 3.83±10.40 | **2.95±0.12** | 1.72±1.34 | **3.29±0.64** |
| R1BNN PAD | **3.84±0.21** | **4.30±0.18** | **3.78±0.18** | 0.09±0.22 | **-3.25±0.20** | 3.13±0.05 | **1.27±0.10** | 4.14±0.03 |
| SWAG | 5.02±2.26 | 4.85±1.00 | 3.73±2.83 | **-0.96±0.18** | -2.96±0.37 | **3.03±0.14** | 1.41±0.39 | 4.25±0.71 |
| SWAG PAD | **3.80±0.58** | **4.47±0.29** | **3.51±1.69** | -0.58±0.09 | -2.46±0.48 | 3.18±0.12 | **1.20±0.09** | **3.53±0.10** |
| MC Drop | 5.37±1.42 | 5.88±1.79 | 4.04±4.23 | **-0.82±0.23** | 4.93±7.55 | **3.09±0.15** | 1.70±0.53 | **3.21±2.05** |
| MC Drop PAD | **4.32±1.92** | **4.98±0.79** | **3.34±0.86** | -0.53±0.09 | **-0.96±3.39** | 3.18±0.08 | **1.22±0.07** | 3.45±0.14 |

Table 2: Calibration error on UCI regression datasets. GP's and FVBNN's are included for reference. **bold** entries contain the best result for the base model. underlined entries are those with a large difference of $\geq 5$ between methods.

| Model | Housing | Concrete | Energy | Kin8nm | Naval | Power | Wine | Yacht |
|---|---|---|---|---|---|---|---|---|
| GP | 8.24±2.73 | 3.34±2.68 | 4.88±3.14 | 2.17±0.95 | 6.65±7.39 | 4.30±4.72 | 1.52±1.07 | 5.10±3.50 |
| FVBNN | 7.10±7.76 | 7.79±5.77 | 5.90±8.62 | 4.46±5.06 | 3.23±2.41 | 2.06±1.47 | 2.10±0.00 | 2.46±0.00 |
| DUN | 15.8±21.5 | 14.3±10.9 | 19.5±23.9 | 16.5±14.0 | 22.4±5.4 | 19.3±20.7 | 18.7±24.4 | 18.9±16.9 |
| DE | 15.95±6.97 | 18.83±8.66 | **5.54±8.91** | **1.44±1.36** | 8.06±5.77 | 2.56±2.53 | 2.81±1.14 | 6.61±6.12 |
| DE PAD | **5.12±3.62** | **12.87±6.66** | 6.34±7.08 | 2.38±1.89 | **6.19±3.93** | **1.60±1.48** | **1.53±1.05** | **2.37±0.18** |
| R1BNN | 6.42±5.45 | 14.18±8.77 | 4.82±9.37 | **1.63±1.57** | 6.41±4.93 | **1.07±1.25** | 2.29±1.50 | 2.82±0.61 |
| R1BNN PAD | **6.04±3.93** | **2.94±3.10** | **4.42±3.83** | 4.63±5.06 | **3.77±2.75** | 1.38±0.61 | **2.14±1.39** | **2.18±0.09** |
| SWAG | 12.43±6.84 | 10.16±6.79 | **5.74±8.85** | 1.62±1.78 | 2.98±2.36 | 1.57±2.19 | 1.85±2.26 | 4.91±1.55 |
| SWAG PAD | **6.73±6.44** | **8.72±4.93** | 6.05±8.60 | 2.15±0.77 | 4.59±1.69 | 1.91±1.87 | **0.74±0.51** | **2.76±0.45** |
| MC Drop | 11.48±6.17 | 14.92±9.90 | **5.98±8.94** | **2.26±2.31** | 5.75±2.88 | 2.33±2.17 | 1.46±0.73 | 6.74±6.95 |
| MC Drop PAD | **8.38±8.32** | **12.78±6.71** | 6.30±7.71 | 2.30±1.29 | **5.52±3.96** | **1.90±1.66** | **1.39±1.11** | **3.58±0.81** |

Classification experiments are done on both MNIST (LeCun et al., 2010) and CIFAR-10 (Krizhevsky et al., 2009) datasets. We use an architecture consisting of 4 and 5 convolutional layers respectively, followed by 3 fully connected layers. We provide extra information regarding the exact architecture in the appendix (section 8). Instead of clustering to create shifted test distributions, we use image corruptions as were used by Snoek et al. (2019). As PAD works in any arbitrary feature space, we apply our method in the latent space between the last convolutional layer and the fully connected layers. We report classification accuracy, negative log likelihood, and expected calibration error (ECE) (Guo et al., 2017) over 5 runs for all models.

## 4.2 BASELINES

We compare PAD against a number of Bayesian models and neural networks including Gaussian Processes, Functional Variational Bayesian Neural Networks (FVBNN) (Sun et al., 2019), Monte Carlo Dropout (MC Drop) (Gal & Ghahramani, 2015), Deep Ensembles (DE) (Lakshminarayanan et al., 2017), SWAG (Maddox et al., 2019), and Rank One Bayesian Neural Networks (R1BNN) (Dusenberry et al., 2020) and Depth Uncertain Networks (DUN) (Antorán et al., 2020).

## 4.3 ANALYSIS

For regression it can be seen in (table 1) that PAD generally improves the likelihood in the scenario where the shifted data distribution caused the baseline model to perform poorest. The underlined

Table 3: Classification accuracy, NLL, and calibration error on OOD test sets for classification. Each row contains a baseline model, with Mixup and PAD variants. **Left**: Accuracy, **Middle**: ECE, **Right**: NLL

| ACCURACY | | | ECE | | | NLL | | |
|---|---|---|---|---|---|---|---|---|
| Model | MNIST | CIFAR-10 | Model | MNIST | CIFAR-10 | Model | MNIST | CIFAR-10 |
| MC Drop | 0.61±0.01 | **0.45±0.01** | MC Drop | 0.31±0.02 | 0.38±0.02 | MC Drop | 2.36±0.26 | 3.23±0.40 |
| MC Drop PAD | **0.62±0.01** | 0.45±0.01 | MC Drop PAD | **0.07±0.07** | **0.05±0.01** | MC Drop PAD | **1.20±0.11** | **1.69±0.02** |
| DE | **0.68±0.00** | **0.52±0.00** | DE | 0.24±0.01 | 0.26±0.01 | DE | 1.52±0.11 | 2.06±0.03 |
| DE PAD | 0.60±0.01 | 0.48±0.01 | DE PAD | **0.18±0.02** | **0.14±0.03** | DE PAD | **1.35±0.07** | **1.69±0.05** |
| R1BNN | 0.69±0.03 | 0.43±0.01 | R1BNN | 0.10±0.05 | **0.06±0.02** | R1BNN | 1.47±0.58 | 1.76±0.01 |
| R1BNN PAD | **0.69±0.04** | **0.45±0.00** | R1BNN PAD | **0.07±0.04** | 0.07±0.04 | R1BNN PAD | **0.98±0.07** | **1.67±0.01** |
| SWAG | 0.68±0.01 | **0.59±0.00** | SWAG | 0.22±0.01 | **0.10±0.00** | SWAG | 1.75±0.18 | **1.28±0.01** |
| SWAG PAD | **0.69±0.01** | 0.59±0.00 | SWAG PAD | **0.11±0.02** | 0.17±0.01 | SWAG PAD | **1.10±0.08** | 1.33±0.02 |

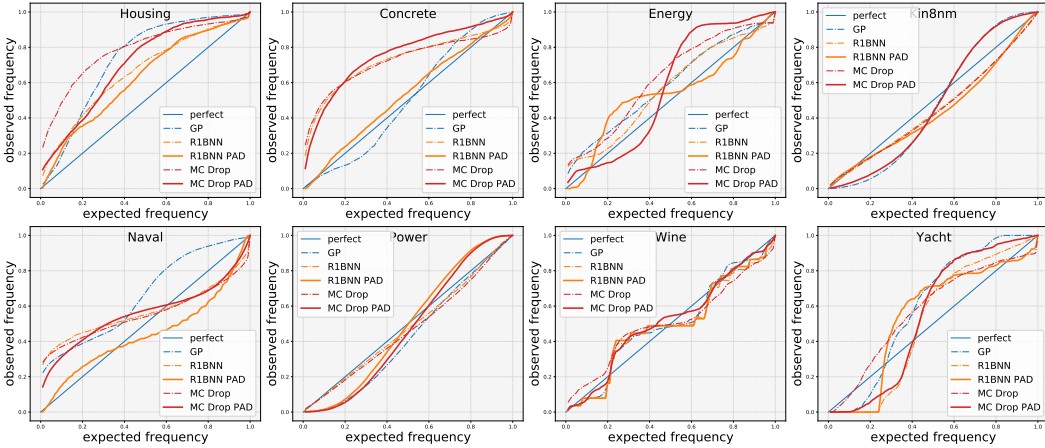

Figure 3: Calibration curves for each model on each dataset. Base models are shown with dotted lines. Our models (model + PAD) are shown with solid lines. GP's are included with a solid line for reference.

entries in (table 1) are those which the negative log likelihood differs by $\geq 1$ on the log scale. It can be seen that the majority (8/9) of these increases in likelihood are achieved by PAD models. In terms of single experiments, PAD beat the baseline models $21/32 \approx 66\%$ of the time. For calibration, in (table 2), the only difference is that the underlined entries are those for which differ by $\geq 5$. For calibration, it can be seen that PAD models contain all cases where a large difference between methods is present. In terms of single experiments, PAD beat the baselines $22/32 \approx 69\%$ of the time.

In (figure 3) we show calibration curves for MC Drop and R1BNN models on each dataset (other models included in figure 9). Baseline models are shown with a dotted line, while PAD models are shown with a solid line. It can be seen that in datasets where the baseline model is poorly calibrated, such as Housing and Concrete, PAD results in curves which are closer to the center line than the baseline models. On datasets where the baseline models tend to be well calibrated, such as Power and Kin8nm, PAD does not result in a large deviation from the baseline. We attribute this to the fact that we tune the lengthscale parameter in such a way that if no improvement can be made over the baseline model on the validation set, the lengthscale parameter $\ell$ and the second term in (6) will have little to no effect on the resulting model parameters. However, the opposite is also true, which leads to large improvements in negative log likelihood and calibration error in the worst case.

It is interesting to note that PAD does not give uniform performance gains across all datasets. To further investigate this, we used TSNE (Hinton & Roweis, 2002) to embed both the natural data $\mathbf{X}$, and the generated data $\tilde{\mathbf{X}}$ for all datasets (figures 6 and 7). Datasets which naturally form a single dense cluster, such as Kin8nm and Power are consistently the lowest performers when paired with PAD, where pad shows the best performance in $1/8$ (NLL) and $2/8$ (calibration) experiments. For datasets which form clusters on manifolds, such as Naval and Yacht, PAD gives the best results on

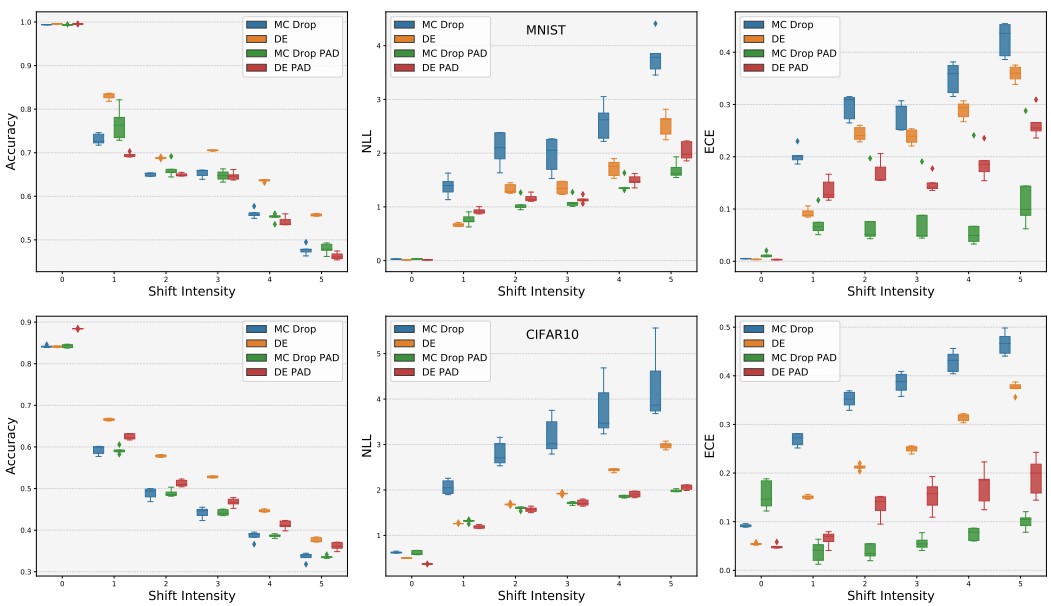

Figure 4: Model performance on varying degrees of shift intensity for MNIST-C and CIFAR10-C. 0 represents the original test set while 6 represents the most extreme level of shift. Models which are augmented with PAD show comparable performance on the natural test set. As shift intensity increases, PAD models exhibit superior performance in terms of negative log likelihood and calibration error. Other models are included in figure 10

4/8 (NLL) and 7/8 (calibration) experiments. PAD shows the strongest performance on datasets which naturally form separable clusters such as Housing, Concrete, Energy, and Wine, for which it performs best on 16/16 (NLL) and 13/16 (calibration) experiments. These results suggest that PAD would be most useful in situations where the data naturally form loose, separable clusters where it is likely that a new cluster may form outside of the region of clusters which are known during training.

## 4.4 ABLATION STUDY

Table 4: Equation 5 Ablation: NLL

| Dataset | Regular | Without A | Without B | Without AB |
|---|---|---|---|---|
| Housing | **4.32±1.92** | 4.62±2.31 | 4.69±2.30 | 4.70±2.31 |
| Concrete | 4.98±0.79 | **4.96±0.73** | 5.05±0.74 | 5.05±0.74 |
| Energy | 3.34±0.86 | **3.26±1.00** | 3.34±1.39 | 3.34±1.39 |
| Kin8nm | **-0.53±0.09** | -0.53±0.09 | -0.48±0.11 | -0.48±0.11 |
| Naval | **-0.96±3.39** | 0.21±3.77 | 0.31±3.69 | 0.22±3.20 |
| Power | **3.18±0.08** | 3.23±0.10 | 3.23±0.20 | 3.23±0.20 |
| Wine | **1.22±0.07** | 1.34±0.57 | 1.72±1.71 | 1.70±1.66 |
| Yacht | 3.45±0.14 | 3.45±0.11 | 3.41±0.13 | **3.41±0.13** |

Table 5: Equation 5 Ablation: Calibration Error

| Dataset | Regular | Without A | Without B | Without AB |
|---|---|---|---|---|
| Housing | **8.38±8.32** | 9.43±8.22 | 9.69±8.35 | 9.70±8.35 |
| Concrete | **12.78±6.71** | 14.04±7.44 | 14.53±7.31 | 14.53±7.31 |
| Energy | **6.30±7.71** | 6.40±7.89 | 6.53±8.29 | 6.52±8.30 |
| Kin8nm | 2.30±1.29 | **2.24±1.43** | 2.47±1.61 | 2.47±1.61 |
| Naval | 5.52±3.96 | **5.44±3.08** | 5.88±3.94 | 5.65±3.71 |
| Power | 1.90±1.66 | 1.97±1.56 | **1.83±1.76** | 1.83±1.82 |
| Wine | 1.39±1.11 | 0.50±0.31 | 0.50±0.31 | **0.48±0.28** |
| Yacht | 3.58±0.81 | **3.42±0.74** | 3.44±0.82 | 3.44±0.82 |

In order to understand the effect of each term in equation 5, we performed an ablation on each term on the MC PAD (MC Dropout + PAD) variant. It can be seen that as more terms are removed from the equation, the model performance degrades in both NLL and calibration. The effect is more pronounced for NLL, as the full equation contains the majority of the best performances.

## 5 RELATED WORK

**Bayesian Methods for Deep Learning**  Functional Bayesian Neural Networks (FVBNN) (Sun et al., 2019) use a functional prior from a fully trained GP (Rasmussen, 2003). This approach is somewhat similar to PAD, but PAD accomplishes this in a way which adversarially generates difficult samples for $f_\theta$ and does not require specifying a GP kernel and then fitting data to it with

the limited expessivity of a GP. (Maddox et al., 2019) recently proposed SWAG, which keeps a moving average of the weights, and subsequently builds a multivariate Gaussian posterior $p(\theta|\mathcal{D})$ which can be sampled for inference. Deep ensembles (Lakshminarayanan et al., 2017) simply trains an ensemble of $N$ independent models, and has become a strong standard baseline in the calibration literature. The resulting ensemble of networks are sampled on new inputs and combined in order to achieve set of highly probable solutions from the posterior $p(\mathbf{w}|\mathcal{D})$. While simple and effective, it can require excessive computation for large models and datasets. (Dusenberry et al., 2020) proposed R1BNN, which uses shared set of parameters $\theta$ along with multiple sets of rank one parameters $\mathbf{r}_i$ and $\mathbf{s}_i$, which can combine via an outer product $\theta \odot \mathbf{rs}^T$ to create a parameter efficient ensemble. Other recent advances include gathering ensemble members by treating depth as a random variable and ensembling the latent features through a shared output layer (Antorán et al., 2020). Recently, there has been resurgence of interest in distance sensitive RBF networks (LeCun et al., 1998) for modeling predictive uncertainty (Liu et al., 2020; van Amersfoort et al., 2020) with both Bayesian and deterministic methods, highlighting the importance of distance in uncertainty modeling.

**Data Augmentation**   The data augmentation literature is vast, and as such we will only mention some of the most relevant works here. Adversarial examples (Goodfellow et al., 2014b) utilize the gradient of the input with respect to the loss which is then used to create a slightly perturbed input which would cause an increase to the loss, and further training with that input. The resulting network should be more robust to such perturbations. (Zhang et al., 2017) proposed Mixup, which creates linear interpolations between natural inputs with the hopes of creating a robust network with linear transitions between classes instead of having a hard decision boundary. Subsequent work by (Thulasidasan et al., 2019) showed that mixup training and the soft decision boundary has the effect of improving network calibration.

**Set Encoding**   Methods which operate as a function of sets have been an active topic in recent years. Deep Sets (Zaheer et al., 2017) first proposed a model $f(\{\mathbf{x}_i, \mathbf{x}_{i+1}, ..., \mathbf{x}_n\}) \rightarrow \mathbb{R}^d$ which passes the input set through a feature extractor, before being aggregated with a permutation invariant pooling function, and then decoded through a DNN which projects the set representation to the output space $\mathbb{R}^d$. (Lee et al., 2019) proposed to use attention layers to create the set transformer. Our method builds on the idea of set based methods by using the sets towards a new objective of identifying and generating data in underspecified regions where the model is likely overconfident.

## 6   CONCLUSION

We have proposed a new method of increasing accuracy and calibration of Bayesian neural network models which we named Prior Augmented Data (PAD). Our method works by encouraging a Bayesian reversion to the prior beliefs of the labels for inputs in previously unseen or sparse regions of the known data distribution. We have demonstrated through various experiments that our method achieves an improvement in likelihood and calibration under shifting data conditions, creating exponential improvements in log likelihood in those conditions which the baseline models tend to make the poorest predictions. One interesting direction of future work could be to investigate a principled way of doing cross validation when training a network with the objective of being robust to OOD data. The OOD data is by definition unknown, so selecting the best hyperparameter settings is not a trivial task. We performed k-fold cross validation using clusters in our regression experiments, but we feel that this topic deserves more study into possible better solutions. Another possible avenue of future research would be to look into the effect that different sizes of $\tilde{\mathbf{X}}$ may have on the resulting uncertainty predictions. It may be the case that the specific cluster sizes and manifold geometries may require more/less attention and therefore some performance and efficiency gains could be had by giving these regsions the proper amount of attention.

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

## 7 APPENDIX

## 8 IMPLEMENTATION DETAILS

### 8.1 $\mathbb{KL}$ DIVERGENCES

For regression tasks we use an analytic calculation of $\mathbb{KL}[p_\theta(y_i|\mathbf{x}_i)||p(y)]$ by assuming a prior of $\mathcal{N}(0, 1)$. In practice, we only want to raise the uncertainty of the prediction while keeping the expressive generalization properties of DNNs, so we use the output of $p_\theta$ as the mean with the standard deviation set to 1.

$$\mathbb{KL}[p_\theta(y_i|\mathbf{x}_i)||p(y)] := \mathbb{KL}[\mathcal{N}_\theta(\mu_i, \sigma_i)||\mathcal{N}_\theta(\mu_i, 1)] \tag{9}$$

For classification tasks, the prior is a uniform categorical distribution. In practice, we found that directly raising the entropy of the output led to a degradation in accuracy, so we instead add another output parameter to the base model such that instead of outputting $C$ class logits, we output $C + 1$ and treat the extra output logit as $\log t$ and use it as a temperature scaling parameter $\sigma(z/t)$ where $\sigma$ is the softmax function. We then implement the $\mathbb{KL}$ divergence by setting a hyperparameter $\tau$ controlling the maximum temperature, and conditionally raising the temperature by the following function.

$$T = (\tau^w - t)^2 \tag{10}$$

where $w$ is the weight given by the exponential term before the $\mathbb{KL}$ divergence in 6. In this way, the minimum temperature of 1 is enforced when $w = 0$, and the maximum temperature of $\tau$ is enforced when $w = 1$. Importantly, we do not transform the logits by when calculating the likelihood loss (first term in equation 8) because we found that this led to a larger generalization error in practice.

### 8.2 CONVOLUTIONAL ARCHITECTURE DETAILS

Table 6: Convolutional architecture used for MNIST experiments

| Layers |
| --- |
| Conv2d(1, 32) $\to$ BatchNorm $\to$ LeakyReLU $\to$ MaxPool $\to$ Dropout(0.05) |
| Conv2d(32, 64) $\to$ BatchNorm $\to$ LeakyReLU $\to$ MaxPool $\to$ Dropout(0.05) |
| Conv2d(64, 128) $\to$ BatchNorm $\to$ LeakyReLU $\to$ MaxPool $\to$ Dropout(0.05) |
| Conv2d(128, 256) $\to$ BatchNorm $\to$ LeakyReLU $\to$ AvgPool $\to$ Dropout(0.05) |
| FC(128, 128) $\to$ ReLU $\to$ Dropout(0.1) |
| FC(128, 128) $\to$ ReLU $\to$ Dropout(0.1) |
| FC(128, 10) |

Table 7: Convolutional architecture used for CIFAR-10 experiments

| Layers |
| --- |
| Conv2d(3, 32) $\to$ BatchNorm $\to$ LeakyReLU $\to$ MaxPool $\to$ Dropout(0.05) |
| Conv2d(32, 64) $\to$ BatchNorm $\to$ LeakyReLU $\to$ MaxPool $\to$ Dropout(0.05) |
| Conv2d(64, 128) $\to$ BatchNorm $\to$ LeakyReLU $\to$ MaxPool $\to$ Dropout(0.05) |
| Conv2d(128, 256) $\to$ BatchNorm $\to$ LeakyReLU $\to$ AvgPool $\to$ Dropout(0.05) |
| Conv2d(256, 256) $\to$ BatchNorm $\to$ LeakyReLU $\to$ AvgPool $\to$ Dropout(0.05) |
| FC(128, 128) $\to$ ReLU $\to$ Dropout(0.1) |
| FC(128, 128) $\to$ ReLU $\to$ Dropout(0.1) |
| FC(128, 10) |

| **Algorithm 1:** PAD for input space | **Algorithm 2:** PAD for latent features |
|---|---|
| **for** $\lvert \mathcal{D}_B \rvert = (\mathbf{X}_B, \mathbf{y}_B)$ **do** | **for** $\lvert \mathbf{X}_B \rvert$ **do** |
| $\quad$ optimize $\theta$... | $\quad$ optimize $\theta$... |
| $\quad \tilde{\mathbf{X}}_B \sim q_\phi(\tilde{\mathbf{X}}_B \lvert \mathbf{X}_B)$. | $\quad \hat{y}_n = f_\theta(\mathbf{x}_n)$ for $n = 1, \ldots, B$. |
| $\quad \hat{y}_n = f_\theta(\mathbf{x}_n)$ for $n = 1, \ldots, B$. | $\quad \hat{\mathbf{z}}_n = f_\theta(\mathbf{x}_{:n})$ for $n = 1, \ldots, B$. |
| $\quad \hat{\tilde{y}}_n = f_\theta(\tilde{\mathbf{x}}_n)$ for $n = 1, \ldots, B$. | $\quad \tilde{\mathbf{Z}}_B \sim q_\phi(\tilde{\mathbf{Z}}_B \lvert \mathbf{Z}_B)$. |
| $\quad \theta_{t+1} \leftarrow \theta_t - \nabla_\theta \mathcal{L}_\theta(\mathcal{D}_B)$ | $\quad \hat{\tilde{y}}_n = f_\theta(\tilde{\mathbf{z}}_{n:})$ for $n = 1, \ldots, B$. |
| $\quad$ optimize $\phi$... | $\quad \theta_{t+1} \leftarrow \theta_t - \nabla_\theta \mathcal{L}_\theta(\mathcal{D}_B)$ |
| $\quad \tilde{\mathbf{X}}_B \sim q_\phi(\tilde{\mathbf{X}}_B \lvert \mathbf{X}_B)$. | $\quad$ optimize $\phi$... |
| $\quad \hat{\tilde{y}}_n = f_\theta(\tilde{\mathbf{x}}_n)$ for $n = 1, \ldots, B$. | $\quad \hat{\mathbf{z}}_n = f_\theta(\mathbf{x}_{:n})$ for $n = 1, \ldots, B$. |
| $\quad \phi_{t+1} \leftarrow \phi_t - \nabla_\phi \mathcal{L}_\phi$; | $\quad \tilde{\mathbf{Z}}_B \sim q_\phi(\tilde{\mathbf{Z}}_B \lvert \mathbf{Z}_B)$. |
| | $\quad \hat{\tilde{y}}_n = f_\theta(\tilde{\mathbf{z}}_{n:})$ for $n = 1, \ldots, B$. |
| | $\quad \phi_{t+1} \leftarrow \phi_t - \nabla_\phi \mathcal{L}_\phi$; |

Figure 5: Training steps for $f_\theta$ and $g_\phi$. PAD can work in both the input space (**left**) and the latent space (**right**)

## 8.3 NEW YORK REAL ESTATE

To apply PAD on a real world problem where the dataset shift is not synthetically created, we applied it to regression on New York real estate data. The dataset consists of 12,000 sales records spanning over 12 years. Each instance has a total of 667 features including real valued and one-hot categorical features. We used the same base models as outlined in section 4. The regression labels are the price that the house sold for. We train the model to predict $\log(y)$ to account for the log-normal distribution of prices and report results based on the log-transformed label. We used the years of 2008-2009 as training/validation data and evaluated the performance on all following years until 2019. It can be seen that with a real temporal distributional shift, PAD models exhibit strong performance. PAD models achieve the best negative log likelihood in 35/40 cases. Similar performance can be seen in terms of calibration error, where PAD models show the best calibration error in 35/40 cases.

Table 8: Negative Log likelihood on 10 years of NY real estate data. The timeline constitutes a real-world temporal dataset shift

| Model | 2010 | 2011 | 2012 | 2013 | 2014 | 2015 | 2016 | 2017 | 2018 | 2019 |
|---|---|---|---|---|---|---|---|---|---|---|
| MC Drop | **-0.08±0.67** | 4.570±12.8 | 36.90±98.3 | 29.55±75.7 | 20.22±51.8 | 9.203±20.6 | 7.050±17.0 | 3.647±7.58 | 0.771±0.97 | 0.856±0.99 |
| MC Drop PAD | 0.549±0.26 | **0.667±0.08** | **1.096±0.11** | **1.167±0.30** | **1.494±0.61** | **1.552±0.59** | **1.251±0.54** | **0.938±0.32** | **0.626±0.23** | **0.654±0.26** |
| DE | **-0.14±0.42** | **0.628±1.00** | 3.973±4.74 | 3.633±3.73 | 4.189±4.24 | 4.148±4.46 | 2.367±2.25 | 1.773±1.57 | 0.750±0.53 | 0.735±0.51 |
| DE PAD | 0.540±0.04 | 0.659±0.06 | **0.971±0.08** | **1.043±0.10** | **1.061±0.21** | **1.217±0.26** | **0.859±0.11** | **0.754±0.07** | **0.656±0.07** | **0.643±0.07** |
| R1BNN | **-0.25±0.05** | **0.367±0.46** | 3.941±2.52 | 4.668±2.66 | 7.047±4.10 | 6.619±5.01 | 5.927±2.97 | 4.548±3.02 | 1.520±1.87 | 1.085±1.66 |
| R1BNN PAD | 0.814±0.01 | 0.855±0.03 | **1.026±0.07** | **1.030±0.06** | **1.020±0.05** | **1.228±0.09** | **0.934±0.02** | **0.893±0.01** | **0.844±0.01** | **0.834±0.01** |
| SWAG | 4.384±3.41 | 4.349±3.44 | 4.426±3.44 | 4.423±3.39 | 4.566±3.41 | 4.781±3.46 | 4.571±3.75 | 4.789±3.55 | 4.818±3.51 | 4.722±3.56 |
| SWAG PAD | **0.811±0.06** | **0.891±0.05** | **1.163±0.10** | **1.254±0.21** | **1.280±0.30** | **1.570±0.26** | **1.148±0.40** | **1.092±0.37** | **1.025±0.36** | **1.006±0.25** |

Table 9: Calibration error across 10 years of NY real estate data. The timeline constitutes a real-world temporal dataset shift.

| Model | 2010 | 2011 | 2012 | 2013 | 2014 | 2015 | 2016 | 2017 | 2018 | 2019 |
|---|---|---|---|---|---|---|---|---|---|---|
| MC Drop | 2.424±1.83 | 7.857±5.87 | 10.34±3.77 | 12.61±4.09 | 10.06±2.05 | 6.270±3.83 | 3.186±2.15 | 3.039±3.20 | 2.318±3.50 | 3.382±6.17 |
| MC Drop PAD | **2.212±1.43** | **2.405±1.25** | **3.580±3.03** | **4.687±3.71** | **5.000±4.31** | **4.076±3.30** | **1.440±1.41** | **1.716±1.63** | **1.149±0.59** | **1.541±0.72** |
| DE | **1.713±0.89** | 5.761±3.11 | 10.61±5.45 | 13.78±6.19 | 11.98±4.45 | 8.402±2.39 | 3.696±1.73 | 5.277±1.68 | 2.855±1.19 | **1.138±0.99** |
| DE PAD | 3.408±0.48 | **3.188±0.68** | **5.131±0.70** | **6.446±1.53** | **6.761±2.40** | **5.260±2.19** | **2.659±1.24** | **3.227±1.15** | **2.757±0.89** | 2.590±0.68 |
| R1BNN | **3.041±3.11** | 7.311±8.33 | 12.17±6.78 | 14.19±5.60 | 12.04±4.62 | 5.771±1.65 | 4.436±1.45 | 4.499±1.36 | **2.387±1.66** | **0.698±0.82** |
| R1BNN PAD | 4.513±0.33 | **3.463±0.41** | **4.080±0.73** | **4.449±0.33** | **4.074±0.42** | **3.255±0.52** | **2.267±0.37** | **2.847±0.34** | 3.301±0.13 | 3.604±0.25 |
| SWAG | 8.207±2.08 | 7.966±2.08 | 7.785±1.92 | 8.100±2.24 | 7.887±2.05 | 7.604±2.32 | 7.883±1.74 | 8.192±1.54 | 8.425±1.09 | 8.350±1.15 |
| SWAG PAD | **3.444±0.51** | **3.201±0.87** | **4.385±1.71** | **5.445±2.21** | **5.146±3.14** | **4.210±2.80** | **2.607±1.90** | **3.209±2.34** | **3.343±1.87** | **3.136±1.27** |

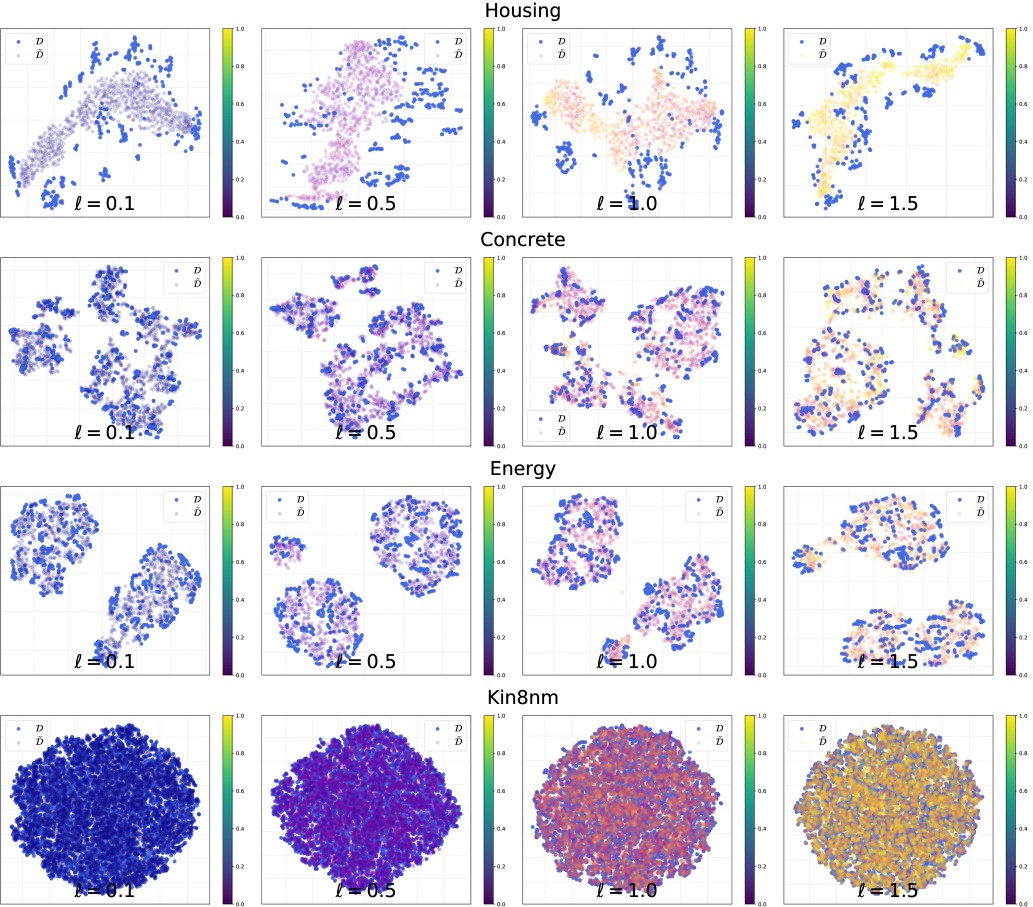

Figure 6: TSNE embeddings for the first half of the UCI regression datatsets we used in our experiments. Both in-distribution $\mathcal{D}$ and out-of-distribution data $\tilde{\mathcal{D}}$ are shown. The color gradient of $\tilde{\mathcal{D}}$ corresponds to different settings for $\ell$ in 5

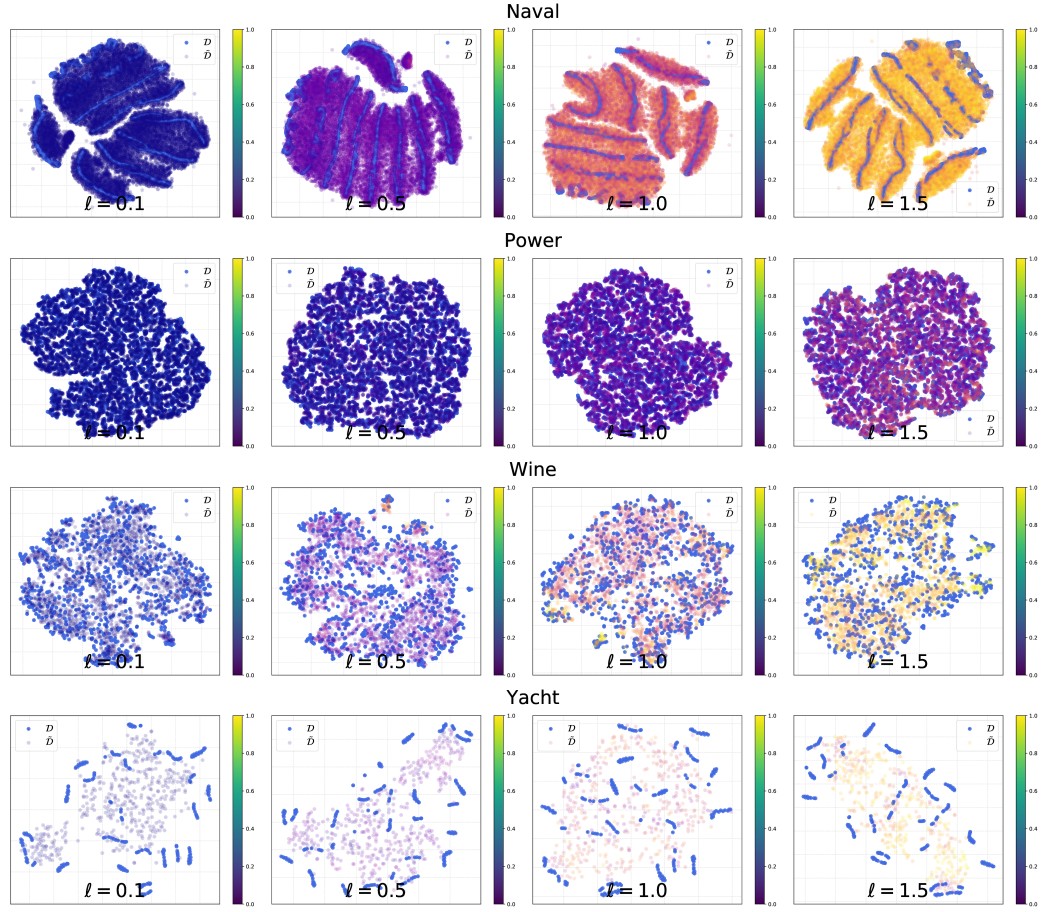

Figure 7: TSNE embeddings for the second half of the UCI regression datatsets we used in our experiments. Both in-distribution $\mathcal{D}$ and out-of-distribution data $\tilde{\mathcal{D}}$ are shown. The color gradient of $\tilde{\mathcal{D}}$ corresponds to different settings for $\ell$ in 5

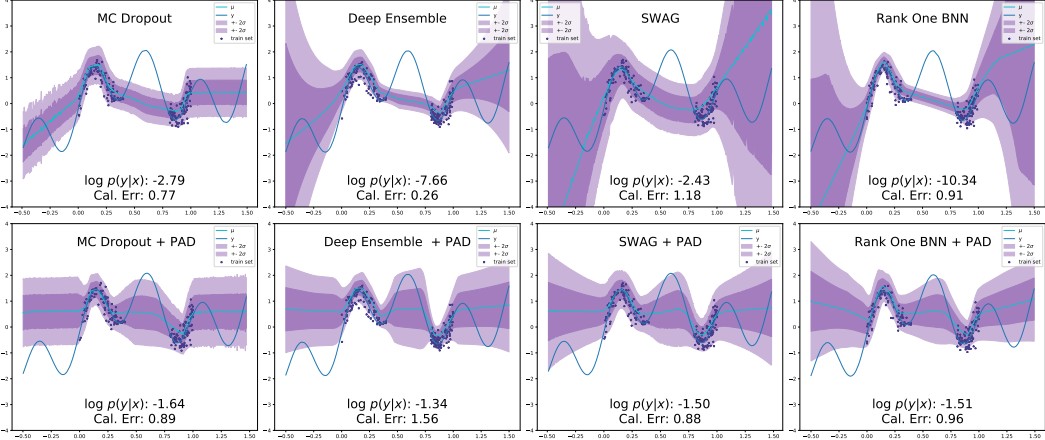

Figure 8: Toy regression experiments on an independent run of the toy experiment. These are included to show the unpredictable behavior of the un-PAD'd models which have an undefined behavior outside of the known regions of training data. Comparing the baselines and PAD with figure 1, it can be seen that the baseline models exhibit unpredictable performance in OOD regions of the input space

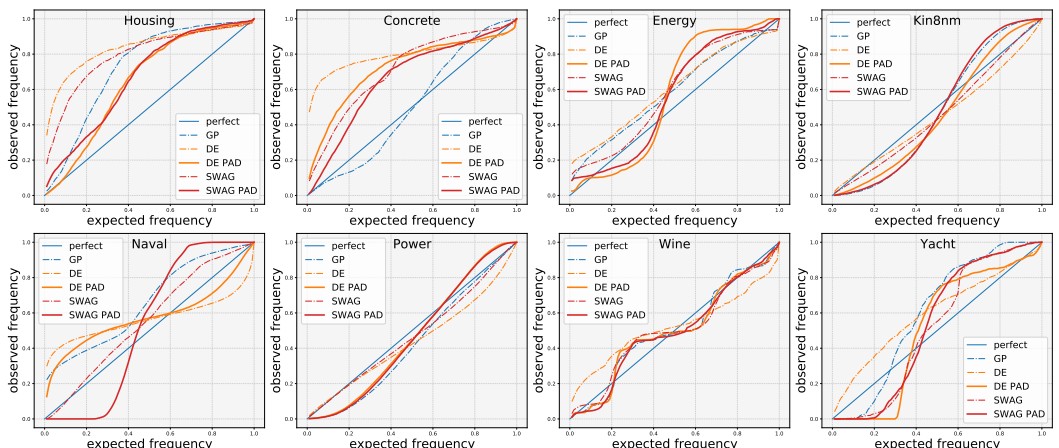

Figure 9: Calibration curves for other models not shown in figure 3.

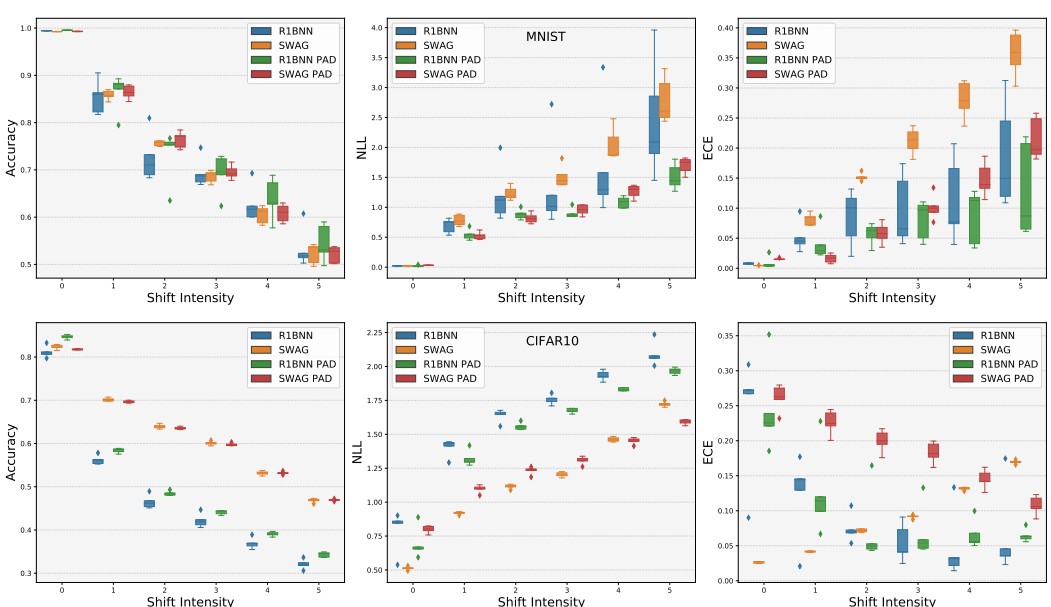

Figure 10: Model performance on varying degrees of shift intensity for MNIST-C and CIFAR10-C. 0 represents the original test set while 6 represents the most extreme level of shift. Models which are augmented with PAD show comparable performance on the natural test set. As shift intensity increases, PAD models exhibit superior performance in terms of negative log likelihood and calibration error. Other models are included in figure 4

