# OpenReview forum: "Improving Neural Network Accuracy and Calibration Under Distributional Shift with Prior Augmented Data "
_ICLR.cc/2021/Conference — Reject_

### Official Review · AnonReviewer1 · 2020-10-19
**The paper improves the calibration on out-of-distribution dataset, by introducing a regularization term using pseudo OOD data generated during training. The proposed method shows improvement in negative log likelihood and calibration error on some dataset in the experiment, where prior is a better approximation than the over-confident baseline metohds.**

**Rating:** 6
**Confidence:** 3

**Review:**

I think the paper is interesting and well-written. I agree with the mis-calibration can be caused by out-of-distribution data, even though it is still commonly observed without such discrepancy. Addressing OOD data is an important direction, and I think the author proposed a reasonable approach to prevent models from overfitting on data points that are rarely observed during training. However, I believe there are some limitation of the method at the current stage, and the experiments did not fully convince me.

Strength:
1. The paper addresses an important question of models being over-confident on out-of-distribution data. The method is practical for applications where uncertainty estimation is needed.
2. The adversarial generation of OOD data is intersting, and the rationale is well explained.
3. The authors included a good selection of datasets and experiments. The PAD-based methods are also compared to a good variety of baselines.

Weakness:
1. To determine whether a data point is out-of-distribution, both the adversarial model and main model rely on the L2 distance and the length scale parameter \ell. My concern here is about 1) the heterogeneity in different dimensions, and 2) \ell seems particularly important and difficult parameter to tune. I would like the authors to give more details on how it is chosen.
2. I believe the approach here is to revert to prior when evaluated data points are far away from the observed ones. Thus the difference in accuracy depends on how good the starting prior is, and how much bias the baseline model learned from OOD data. Table 3 shows some of the trade-off, but I think it also be good to show the difference when there's no OOD data, because it's not necessarily known in advance.
3. Looking at figure 3, I don't think the PAD method shows significant improvement in most of the datasets. Housing seems to be the only one here. In figure 5, it looks like the OOD data are mostly in the convex hull of observed data (at least in this low dimensional embedding). It is unclear how to differentiate those from the region where models should be interpolating. Moreover, all the OOD data are artificially constructed. I think it would be more convincing to test the methods on some OOD data that arises naturally. One such source could be temporal data where distribution could shift over time.

Minor comments:
1. In section 5 "excessive computation for large models an datasets", an->and.

-----------

Thank the authors for lot of these responses. I'm still around neutral for this paper, but I will raise my score to marginally above acceptance.

---

> ### Author Response · Authors · 2020-11-14
> **Added additional experiments on in distribution data and different corruption intensities**
>
> Thank you for taking the time to review our work. We would like to address your concerns in order below.
>
> 1. “Heterogeneity in different dimensions…” We interpret this concern as questioning whether the L2 distance is applicable to feature spaces which are non Euclidian. This coupled with the fact that some feature spaces may be too large, such as images, is why we made the MNIST/CIFAR experiments work in the latent space of the model for which L2 distance is feasible. If we have misinterpreted the purpose of your question, please let us know and we will revisit this point.
>
> ---
>
> 2. "$\ell$ seems particularly important and difficult to tune. I would like more details on how it is chosen." For tuning the $\ell$ parameter on PAD regression models, we mention in section 4.1 that we do k-fold cross validation with $k = 2$, where we cluster the regression dataset and select half of the clusters for each fold. We then tune the $\ell$ parameter along with all other hyperparameters such as dropout rate, learn rate, etc. For classification experiments, there we simply choose a random 10% of the training set as the validation set and tuned $\ell$ as any other hyperparameter.
>
> ---
>
> 3. You are correct in noting that $\ell$ is hard to tune, because it is dependent on the OOD data, which we do not know at train time. We were intrigued as to if there was a better way to do tune this as well, and it was included as possible areas of future research in section 6.
>
> ---
>
> 4. "I think it also be good to show the difference when there's no OOD data..." **We have added an experiment on the regular test set along with increasing levels of corruption, please see figure 4.**
>
> ---
>
> 5. "Looking at figure 3, I don't think the PAD method shows significant improvement in most of the datasets." **We have included a less noisy figure 3, and included the original figure 3 in the appendix. (figure 8)** We hope this better illustrates the performance of PAD. We think it is important to note that the **data displayed in table 2 quantifies the same information as figure 3,** so this should be referenced as well when looking at calibration performance. Figure 3 may visually miss some of the more extreme calibration error which happens near edges of 0 and 100 on the CDF plots.
>
> ---
>
> 6. “One such source could be temporal data where distribution could shift over time” **Updated (11/23): We have added a real world experiment.**
>
> ---
>
> 7. We fixed the typo in section 5 an → and.
>
> We thank you for your constructive review, and have incorporated the relevant changes as mentioned above.

---

> > ### Comment · AnonReviewer1 · 2020-11-24
> > **Thanks for the response**
> >
> > Thank you for the detailed response. I will reconsider my score based on your feedback and discussion with other reviewers.
> >
> > Just to clarify on the comment on the heterogeneity, I was thinking that features might have intrinsically different scale that having different \ell for each dimension (something like ARD kernel) makes more sense. I think a well-chosen embedding space is a reasonable alternative, which I might have missed earlier.

---

> > > ### Author Response · Authors · 2020-11-24
> > > **Thank you for evaluating our response**
> > >
> > > We see your point about $\ell$ on each feature dimension. As you correctly pointed out, we opted to choose using a standard embedding space either from normalized input features or latent embeddings, and use a single scalar $\ell$ parameter.
> > >
> > > Thank you for your continued evaluation and clarification. If there are anything remaining which is not fully addressed in our response, please do not hesitate to ask us for clarification.
> > >
> > > Thank you,
> > >
> > > Authors

---

> ### Author Response · Authors · 2020-11-23
> **Completed real world experiment with temporal shift; Summary of response**
>
> We have completed another experiment on real world data, analyzing PAD's performance on a temporal data shift. The data set we used is from New York City real estate sales prices from 2008-2019.
>
> We used the same base models as outlined in experiments section. Each instance contains 667 features including real values and one-hot categorical features. The regression labels are the price that the house sold for. Since prices are log-normally distributed, we train the model to predict $\log(y)$ and report numbers and report results based on the log-transformed label. We used the years of 2008-2009 as training/validation data and evaluated the performance on all following years until 2019. It can be seen that with a real distributional shift, baseline models still exhibit worse performance in most cases. PAD models show the best negative log likelihood in 35/40 cases. Similar performance can be seen in terms of calibration error, where PAD models show the best calibration error in 35/40 cases.
>
> We have also included these tables in the appendix in section 8.3 (table 8 and 9) where we have included the variance of each measurement. It can be seen that models with PAD exhibit lower variance in addition to better performance.
>
> # Negative Log Likelihood
>
> | Model     | 2010  | 2011  | 2012  | 2013  | 2014  | 2015 | 2016 | 2017 | 2018 | 2019 |
> |-----------|-------|-------|-------|-------|-------|------|------|------|------|------|
> | MC Drop   | **-0.08** | 4.57  | 36.90 | 29.55 | 20.22 | 9.20 | 7.05 | 3.64 | 0.77 | 0.85 |
> | MC PAD    | 0.54  | **0.66**  | **1.09**  | **1.16**  | **1.49**  | **1.55** | **1.25** | **0.93** | **0.62** | **0.65** |
> | DE        | **-0.14** | **0.62**  | 3.97  | 3.63  | 4.18  | 4.14 | 2.36 | 1.77 | 0.75 | 0.73 |
> | DE PAD    | 0.54  | 0.65  | **0.97**  | **1.04**  | **1.06**  | **1.21** | **0.85** | **0.75** | **0.65** | **0.64** |
> | R1BNN     |**-0.25** | **0.36**  | 3.94  | 4.68  | 7.04  | 6.61 | 5.92 | 4.54 | 1.52 | 1.08 |
> | R1BNN PAD | 0.81 | 0.85 | **1.02**  | **1.03**  | **1.02**  | **1.22** | **0.93** | **0.89** | **0.84** | **0.83** |
> | SWAG      | 4.38  | 4.34  | 4.42  | 4.42  | 4.56  | 4.78 | 4.57 | 4.78 | 4.81 | 4.72 |
> | SWAG PAD  | **0.81**  | **0.89**  | **1.16**  | **1.25**  | **1.28**  | **1.57** | **1.14** | **1.09** | **1.02** | **1.00** |
>
> # Calibration Error
>
> | Model     | 2010  | 2011 | 2012 | 2013 | 2014 | 2015 | 2016 | 2017 | 2018 | 2019 |
> |-----------|-------|------|------|------|------|------|------|------|------|------|
> | MC Drop   | -2.42 | 7.85 | 10.3 | 12.6 | 10.0 | 6.27 | 3.18 | 3.03 | 2.31 | 3.38 |
> | MC PAD    | **2.21**  | **2.40** | **3.58** | **4.68** | **5.00** | **4.07** | **1.44** | **1.71** | **1.14** | **1.54** |
> | DE        | **1.71**  | 5.76 | 10.6 | 13.7 | 11.9 | 8.40 | 3.69 | 5.27 | 2.85 | **1.13** |
> | DE PAD    | 3.40  | **3.18** | **5.13** | **6.44** | **6.76** | **5.26** | **2.65** | **3.22** | **2.75** | 2.59 |
> | R1BNN     | **3.04**  | 7.31 | 12.1 | 14.1 | 12.0 | 5.77 | 4.43 | 4.49 | **2.38** | **0.69** |
> | R1BNN PAD | 4.51  | **3.46** | **4.08** | **4.44** | **4.07** | **3.25** | **2.26** | **2.84** | 3.30 | 3.60 |
> | SWAG      | 8.20  | 7.96 | 7.78 | 8.10 | 7.88 | 7.60 | 7.88 | 8.19 | 8.42 | 8.35 |
> | SWAG PAD  | **3.44**  | **3.20** | **4.38** | **5.44** | **5.14** | **4.21** | **2.60** | **3.20** | **3.34** | **3.13** |
>
> In addition, we have also added additional results studying the effect of PAD on different corruption intensities (figure 4).
>
> We have done our best to address every comment in your feedback. The end of the author discussion period is approaching, so if there are any further questions or concerns we would like to do our best to address them as soon as possible.
>
> Thank you for your feedback.

---

### Official Review · AnonReviewer3 · 2020-10-27
**Would like to see a stronger conceptual motivation and defense of the objective function.**

**Rating:** 5
**Confidence:** 3

**Review:**

Overview:

The authors propose a data augmentation scheme that generates samples out of distribution and helps with uncertainty estimates. Comparisons are to various bayesian methods in uci regression and mnist/cfar for classification. PAD seems to give some improvements in out of distribution uncertainty quantification.

The major concern is that the gains seem relatively small, and the objective is ad-hoc. It would be nice to see either more substantial, uniform gains (so that the authors can justify the procedure on the results alone) or more solid conceptual motivation of the method, especially from the Bayesian side. It seems like the motivation and intro is clear, and section 3 onwards becomes very ad-hoc and loses much of this. It would be nice to be convinced that there are a set of assumptions and conditions under which this is the right way to do uncertainty quantification.

Positives:

The evaluations are extensive, and it's commendable that they include both positive and negative results in their regression evaluations.

Uncertainty estimation out of distribution is an important and timely problem.

Negatives:

Minor: I'm not sure why there is a claim that the problems with uncertainty estimation comes from p(theta|D) and not p_theta(y|x). The fact that non-bayesian methods have similar issues with uncertainty quantification would suggest that the latter is certainly an issue.

Figure 1 doesn't seem like a compelling argument for the narrative in the paper. MC dropout and deep ensembles both have decent behavior outside the support (x<0, x>1) but suffer in the gap between 0.25 to 0.75 which is arguably due to overaggressive interpolation.

Term A in equation 5 is justified as "generating data where f is overconfident" but I don't see how this is true. It's just generating data where there's low prediction entropy... this includes areas where it is confident for the right reasons.

Somewhat minor, but the sum of term A and term B seems a bit problematic, since A will be in terms of discrete entropy in classification, and term B is going to be differential entropy in general. Rescaling X also seems like it would arbitrarily shift the weights between AB and C? The weird thresholding on the C penalty for regression problems does not inspire confidence.

Overall, equation 5 gives a sense of a fairly ad-hoc criterion. I'd like to be convinced that this is actually the right way of doing things, especially from a Bayesian perspective.

Looking at equation (7) it seems like learning the distribution of tilde X is alot of work to regularize the KL towards the marginal with a squared-exponential penalty away from the training data. Is it really not possible to post-process the model distribution to achieve the same thing?

The experiments are extensive, but a bit mixed. The dataset construction for regression seems like it would naturally favor PAD-type methods, because the clustering occurs on the basis of feature distances, and PAD enforces uncertainty based on feature distances (via the squared-exp term in equation 7). In terms of results, I think overall PAD gives gains, but it's not uniform and in cases like SWAG on table 2 seems to hurt more than it helps.

The corruptions in MNIST / CIFAR must also be pretty aggressive, as the accuracy numbers are quite low for both. Does PAD do similarly well on milder or no distribution shift settings? I am slightly concerned that the evaluations here focus so much on the large distribution setting and that PAD is tuned to that case.

Minor:

Inline equation involving sines is missing a closing parenthesis.

Notation. g_phi is a 'generative model' in section 2 but it seems to be the output of an autoencoder in section 3.1 q_phi seems to be the actual generative model?

---

> ### Author Response · Authors · 2020-11-14
> **Expanded analysis, added further empirical experiments of objective on different corruption intensities (1/2)**
>
> Thank you for your thoughtful comments. We will address the issues raised below in order.
>
> 1. “The major concern is that the gains seem relatively small” We disagree that the gains are small. **We have underlined cases in the results tables where the gains in NLL and Calibration achieved by PAD’d models are extreme.** We have added extra analysis to section 4.3 as to why the gains may not be expected to be uniform for all datasets.
>
> ---
>
> 2. "I'm not sure why there is a claim that the problems with uncertainty estimation comes from $p(\theta|D)$..."It may be possible to focus on only $p(y|x)$, but this treatment only captures aleatoric uncertainty present in the data. In order to effectively model what is unknown, epistemic uncertainties must be taken into account, which necessitates uncertainties over the model parameters. **Therefore, we follow the Bayesian approach of modeling $p(\theta | D)$ because it allows for modeling of epistemic uncertainties.**
>
> ---
>
> 3. “Figure 1 doesn't seem like a compelling argument for the narrative in the paper...”  **figure 1 was replaced (original added to appendix as figure 7)**. We believe it does make a compelling argument for PAD. **MC dropout appears to follow the trend, but that is a random occurrence**, as can be seen by the randomness of the mean predictions in the top row in figure 1, and between independent runs in figures 1 and 7. Also, MC dropout shows a near constant level of uncertainty outside of the data region. Models in the top row have unpredictable mean/variance predictions outside of the data region, and the uncertainty often explodes (outer boundary only) which has a negative effect on the overall likelihood as can be seen in the metrics in figure 1. **Conversely, PAD models exhibit predictable behavior, revert to the prior, and have better log likelihood metrics in both figures 1 and 8.**
>
> ---
>
> 4. “Term A in equation 5 is justified as "generating data where f is overconfident" but I don't see how this is true...” **We have reworded the sentence to use “may be” instead of “is”.** The A term alone seeks out the low entropy areas, but when coupled with the fact that the C term generates the “between” samples, and the $\theta$ update has a coefficient dependent on distance, PAD only updates overconfident areas, with a smooth transition from appropriately confident areas to likely overconfident OOD areas.
>
> ---
>
> 5. "The sum of term A and term B seems a bit problematic..." They will be respectively discrete/differential in classification tasks, but we do not think these objectives interfere with in any meaningful way. A seeks low predictive entropy from $f_\theta$, and B seeks to avoid homogeneous samples from $g_\phi$. We don't follow your meaning about rescaling X and shifting weights, can you please clarify?
>
> ---
>
> 6. "equation 5 gives a sense of a fairly ad-hoc criterion..." Equation 5 arises from a simple and effective solution to the problem of confidence calibration in BNN's [3] as shown in the toy example. We feel we have shown empirically that it achieves its desired objective through a range of experiments in both regression and classification domains.
>
> ---
>
> 7. “Is it really not possible to post-process the distribution to achieve the same thing...” We initially considered post-processing, but it comes with side effects such as the need to train an auxiliary model or augment the inference procedure with training data which both result in increased inference complexity. Therefore, we decided to make PAD part of the training procedure. Moreover, [2] showed previously developed post-processing techniques only improve global calibration, which cannot focus on specific regions or at the instance level. PAD can focus on specific regions of feature space.
>
> ---
>
> 8. “dataset construction for regression seems like it would favor PAD-type methods...” We feel that the regression data were constructed in a realistic way. Real-world scenarios can have missing regions where data gathering is difficult, or temporal shifts which cause new clusters to appear.  **We have also included classification results on datasets proposed by [1] (non-clustering approach). PAD still fared well in these experiments.**
>
> ---
>
> 9. "The experiments are extensive, but a bit mixed." We have added extra analysis and summary statistics to section 4.3 which may address some concerns about when and where mixed results appear.
>
> ---
>
> [1] Hendrycks, Dan, and Thomas Dietterich. "Benchmarking neural network robustness to common corruptions and perturbations." arXiv preprint arXiv:1903.12261 (2019).
>
> [2] Zhao, S., Ma, T., & Ermon, S. (2020). Individual Calibration with Randomized Forecasting. arXiv preprint arXiv:2006.10288.
>
> [3] Ovadia, Y., Fertig, E., Ren, J., Nado, Z., Sculley, D., Nowozin, S., ... & Snoek, J. (2019). Can you trust your model's uncertainty? Evaluating predictive uncertainty under dataset shift. In Advances in Neural Information Processing Systems (pp. 13991-14002).

---

> > ### Author Response · Authors · 2020-11-18
> > **Expanded analysis, added further empirical experiments of objective on different corruption intensities (2/2)**
> >
> >
> > 10.  “Does PAD do similarly well on milder or no distribution shift settings?...” **Our MNIST/CIFAR corruptions are based on [1], which gives 5 corruption intensities, we have included results which take the original test set and differing corruption levels into account, please see figure 4.**
> >
> > ---
> >
> > 11. Closing parentheses in inline equation in section 2.2 → fixed
> >
> > ---
> >
> > 12. "...q_phi seems to be the actual generative model?" $g_\phi$ is a generator which outputs the parameters of a distribution for $\mathbf{\tilde{X}}$, so therefore we can say that $g_\phi$ is a generative model that outputs the distribution $q_\phi$
> >
> > Thank you for constructive feedback, we have done our best to address your concerns and incorporate your advice.

---

> > > ### Comment · AnonReviewer3 · 2020-11-24
> > > **Thanks for the updates.**
> > >
> > > I've read the rebuttal, and appreciate the added experimental validation and results. It gives me more confidence about the empirical aspects of the work, although I continue to find the approach to be ad-hoc (as the rebuttal fails to clarify the conceptual motivation behind the approach, resorting to generic comments about Bayesian uncertainty quantification or wording changes).

---

> > > > ### Author Response · Authors · 2020-11-25
> > > > **On the conceptual motivation**
> > > >
> > > > Thank you for continued evaluation. We would like to note that we included another result in tables 1 and 2 just now comparing to DUN [1], which serves as further evidence of the effectiveness of our approach.
> > > >
> > > > Regarding the conceptual motivation of the objective function. It arises as a novel approach which solves the problem of overconfidence in BNN's. We are unsure about where our conceptual motivation is not clear, so we will provide a brief end-to-end motivation below:
> > > >
> > > > 1. BNN's introduce epistemic uncertainty in the weights.
> > > >
> > > > 2. Because of this epistemic uncertainty, BNN's should not be overconfident where there is a lack of evidence. As
> > > >  illustrated in figure 1, quantified in [2] and our baseline results, BNN's can still be overconfident without evidence.
> > > >
> > > > 3. Therefore, we aim to build on existing BNN's, providing data in local regions of $\mathcal{D}$ where existing evidence is likely sparse, labeling this data with the label prior (which makes $\mathcal{\tilde{D}}$), and using it as a confidence regularizer, solving the problem in #2.
> > > >
> > > > 4. To make $\mathcal{\tilde{D}}$, we use $g_\phi$. We only know $\mathcal{D}$ and don't know where new OOD data will show up. Therefore, the desirable qualities of $\mathcal{\tilde{D}}$ are to be:
> > > >
> > > >     - Where $f_\theta$ shows high confidence (A in eq. 5)
> > > >     - Diverse (B in eq. 5)
> > > >     - In the local vicinity of $\mathcal{D}$. (C in eq. 5)
> > > >
> > > > 5. All that is left is to specify how $f_\theta$ handles $\mathcal{\tilde{D}}$. We view it as a confidence regularization. As the output from $g_\phi$ is diverse, we don't wish to unnecessarily penalize $f_\theta$, and therefore add a coefficient controlled by $\ell$ which decays to 0 as $|| \mathcal{D} - \mathcal{\tilde{D}} ||_2 \rightarrow 0$
> > > >
> > > > Conceptually, this should result in data which fill in spatial gaps in $\mathcal{D}$ for which the regularization strength becomes stronger towards the label prior as the distance from $\mathcal{D}$ increases.  We have pictured this, along with the regularization gradient incurred by different values of $\ell$ in figures 6 and 7.
> > > >
> > > > Throughout our experiments and figures, we have done our best to demonstrate that PAD fulfills our desired objective as intended. While we understand your concerns about the objective being ad-hoc, **we have demonstrated that**:
> > > >
> > > > 1. Even the most recent BNN models suffer from overconfidence.
> > > > 2. PAD delivers an effective solution to this problem which improves performance on OOD data in regression and classification datasets, as well as real-world experiments.
> > > >
> > > > We have done our best to address your concerns, and if there is anything which is still unclear about our motivation, please do not hesitate to ask for clarification.
> > > >
> > > > Thank you,
> > > >
> > > > Authors
> > > >
> > > > [1] https://arxiv.org/abs/2006.08437
> > > >
> > > > [2] Ovadia, Y., Fertig, E., Ren, J., Nado, Z., Sculley, D., Nowozin, S., ... & Snoek, J. (2019). Can you trust your model's uncertainty? Evaluating predictive uncertainty under dataset shift. In Advances in Neural Information Processing Systems (pp. 13991-14002).

---

> ### Author Response · Authors · 2020-11-23
> **Summary of response and changes**
>
> This is a summary of what we have provided in response to your feedback.
>
> - We have reconfigured figure 1, and clarified some misconceptions about what it shows.
> - We have added extra analysis to section 4.3
> - We have added an additional results studying the effect of PAD on different corruption intensities (figure 4)
>
> We have done our best to address every comment in the initial feedback. The end of the author discussion period is approaching, so if there are any further questions or concerns we would like to do our best to address them as soon as possible.
>
> Thank you for your feedback.

---

### Official Review · AnonReviewer4 · 2020-10-29
**A heuristic approach with no guarantees on performance**

**Rating:** 3
**Confidence:** 4

**Review:**

This paper proposes a data augmentation scheme (named PAD) to improve accuracy and calibration of NNs. The idea is to generate OOD data, close to the training data, where the model is overconfident, and force a higher entropy for their corresponding predictions.

This topic is very relevant to the ICLR community, the paper is clear, and I was excited with the goal in a first place. However, the paper as it is has major drawbacks.

1. The biggest drawback is that the proposed approach is ad-hoc, a heuristic with no guarantees that it will work as desired. In fact, recent work has shown that Data augmentation on top of Ensembles can be harmful, the authors should discuss this in the paper (see [Wen et.al, 2020: Combining Ensembles and Data Augmentation can Harm your Calibration). For this paper to be accepted, the authors should explore the properties of the proposed approach with careful controlled toy scenarios, and bring further insights on when the approach is expected (ideally, guaranteed) to work.
2. Using PAD on top of other probabilistic approaches destroys the probabilistic interpretation.
3. Experimental results are extensive, but not convincing: Figure 1 lacks the GP reference, and shows bad performance on the left extreme; the Ablation study suggest that Equation (5) could be simplified; finally results in Table 2-4 suggest that the proposed approach hurts in high-dimensional scenarios (Energy and Kin8nm datasets), the reported numbers also strongly depend on model selection and tuning of PAD and other baselines, information which is currently missing.
4. The authors do not compare nor mention recent advances on calibrating DNNs, for example:
	* (Antoran et.al, 2020) Depth Uncertainty in Neural Networks
	* (Liu et.al, 2019) Simple and principled uncertainty estimation with deterministic deep learning via distance awareness


More comments:

* the proposed model does not seem to scale to high-dimensions, as "filling the gaps" with the OOD data generator becomes infeasible (this is reflected in the tables, where both accuracy and calibration are systematically worse for the high-dim Kin8nm dataset). Up to how many dimensions would this approach be useful?

* the OOD dataset produce an "equally sized pseudo dataset". Yet, one might think that the amount of data needed to robustify uncertainty would depend on the manifold geometry.
* location of OOD samples is chosen as an interpolation of latent representations for the observed data. That means that many generated datapoints will NOT bee out-of-sample.

* From the ablation study (Tables 4 and 5), "without AB" gives similar results to Regular (always within the reported error bars of "regular". That seems to indicate that terms A and B are not that relevant. Am I missing something?

* Figure 1: the authors should include one column for the GP behavior, since the authors claim that the observed behavior is similar to that. Otherwise, it is unclear by eye what is best. In particular, PAD

* Could the proposed approach suffer from the opposite issue, i.e., deliver too high uncertainty in the augmented OOD data? How do you avoid this issue?
* How does the proposed approach compare to a DNN whose last layer is GP or Bayesian RBF network? (see http://www.gatsby.ucl.ac.uk/~balaji/udl2020/accepted-papers/UDL2020-paper-009.pdf)
* The proposed method encourages a reversion to a *specific* prior (0 mean functions)

Minor:

* the authors mention limited expressiveness of GPs, but this is subject to a simple kernel. If the kernel is complicated enough, then GPs are as expressive as we would like to (see equivalences between DNN and GPs in [Neil, 1997] and [Lee et.al, 2017]). Please clarify this statement.
* Figure 3 is hard to read, I suggest to highlight the PAD curves by changing the color scheme).

---

> ### Author Response · Authors · 2020-11-14
> **Added expanded analysis, approach empirically shown to work on a variety of tasks (1/2)**
>
> Thank you for your review and comments. We will address your concerns in order below...
>
> 1. "The biggest drawback is that the proposed approach is ad-hoc..." We agree the approach may be called ad-hoc, but **it arises as a simple and effective solution to a known problem in BNN's [2], overconfidence without justification.** We have shown the empirical effectiveness of our method through toy examples and experiments in both regression and classification domains. **Likewise, we have added further analysis to section 4.3 indicating when PAD has been empirically shown to be most effective**
>
> ---
>
> 2. "...recent work has shown Data augmentation on top of Ensembles can be harmful." **[1] is a concurrent ICLR 2021 submission**. [1] shows data augmentation combined with ensembles produces abnormally low confidence, and fix this by a heuristic which only applies mixup to overconfident classes. PAD shares similarities with the heuristic in [1], such as 1) identifying where the model(s) need confidence adjustment, and applying said adjustment. In PAD, the necessary condition and adjustment is based upon the empirical features and the model being trained instead of a validation heuristic. **A key consequence of the distance based metric of PAD is that it can be extended to the regression case which is often overlooked within the calibration literature which often focuses on classification tasks, as [1] does.** To show that PAD does not make ensemble methods underconfident, **we have included another experiment in figure 4, which shows that when PAD is added to different ensemble methods, it shows better OOD calibration, and does not suffer from underconfidence.**
>
> ---
>
> 3. "Using PAD on top of other probabilistic approaches destroys the probabilistic interpretation." PAD generates realistic instances which are no different from natural data from the point of view of the base model. Therefore, the probabilistic interpretation of the base model remains intact.
>
> ---
>
> 4. "Figure 1 lacks the GP reference, and shows bad performance on the left extreme." **We added a GP for reference in figure 1.** We also added the original figure to the appendix (figure 8), as a cross reference. We disagree that the left of figure 1 exhibits bad performance. The baseline models' performance around the outer boundaries of the training data is random between models/runs (see difference between the top rows of figure 1 and 8). **Different runs of the toy example yield different results for OOD mean and variance behavior. On the other hand PAD models show more predictable behavior around the boundaries and in regions of missing data, much like the GP. Most importantly, the log likelihood metrics for PAD are much better than the baseline models in figure 1 because the uncertainty does not unpredictably explode, but reverts towards the prior.**
>
> ---
>
> 5. “...ablation suggests Equation (5) could be simplified...” We read the table differently. For NLL, 62.5% of the best results are in the column with the full equations given. The other two columns with bold items contain 20% and 12.5% of the best results respectively. In every case where the full equation is not the best, the maximum difference is at most $0.08$ whereas the maximum gain by using the full equation is at most 1.17, indicating that there is a large upside and limited downside for NLL. We agree that the ablation study becomes less clear in the case of calibration, but we include it for completeness and believe that calibration results in table 2 speak for themselves.
>
> ---
>
> 6. “The reported numbers also strongly depend on model selection and tuning of PAD and other baselines, information which is currently missing...” **We do include the tuning procedure for all models in section 4.1.**
>
> ---
>
> 7. “... the proposed approach hurts in high-dimensional scenarios (Energy and Kin8nm datasets)...” Scaling to high dimensional data was a focus of ours. it is important to note that the dimensionality of the **Kin8nm/Energy datasets were not an issue, as they both have 8 dimensions, while Boston-Housing has 13,** the latter being a very strong performing dataset under PAD. In addition to low dimensional regression experiments, **we also included classification experiments which operate on a 128 dimensional latent space before the FC layer, and PAD still showed strong performance.** We have added extra analysis to section 4.3 about the Kin8nm/Energy datasets.
>
> ---
>
>
> [1] https://openreview.net/forum?id=g11CZSghXyY
>
> [2] Ovadia, Y., Fertig, E., Ren, J., Nado, Z., Sculley, D., Nowozin, S., ... & Snoek, J. (2019). Can you trust your model's uncertainty? Evaluating predictive uncertainty under dataset shift. In Advances in Neural Information Processing Systems (pp. 13991-14002).

---

> > ### Author Response · Authors · 2020-11-14
> > **Added expanded analysis, approach empirically shown to work on a variety of tasks (2/2)**
> >
> > 8. “Could the proposed approach suffer from the opposite issue...” **We prevent this via the limiting prior and the $1 - \exp(...)$ term in equation (7).** If the distance to the closest point in the current batch approaches 0, then the exponential term in (7) will go to zero, while it will approach one as that minimum distance increases. **Coupled with the tuning of $\ell$ on a validation set in section 4.1, we can be reasonably certain that we will not raise the uncertainty without justification.**
> >
> > ---
> >
> > 9. "amount of data needed to robustify uncertainty would depend on the manifold geometry." This is an interesting idea, and something we will consider for future research. Dependently changing the generated dataset size is itself a novel and complex problem, and we feel that could be the entire basis of an entirely new work. **This idea has been added to section 6.**
> >
> > ---
> >
> > 10. “OOD samples is chosen as an interpolation of latent representations...” This depends on how OOD is viewed. $g_\phi$'s objective is to generate samples where $f_\theta$ is confident and where an existing $x_i$ does not exist. In this sense, the $x_i$ has not been seen in $X$ is therefore OOD. **There is no strict interpolation as $g_\phi$ is free to generate data according to its objective**, and the ultimate goal is to **provide a transition from confident in distribution data to uncertain OOD data** which means that the $\tilde{x_i}$ lies in the transition between in distribution and OOD. We therefore classify $\tilde{X}$ as OOD data.
> >
> > ---
> >
> > 11. "The authors do not compare nor mention recent advances on calibrating DNNs..." The SWAG and R1BNN baselines are two recent models which aim to calibrate BNN's and we have provided a total of 4 BNN baselines which we apply PAD to. The models proposed in your comment are interesting works, and we have added mentions to them in the related work. **We will consider adding them as baselines subject to the completion of other experiments in the rebuttal period.**
> >
> > ---
> >
> > 12. "The authors mention limited expressiveness of GPs, but this is subject to a simple kernel." **We have updated the statement in the paper to specify GP’s with an RBF kernel.**
> >
> > ---
> >
> > 13. "The proposed method encourages a reversion to a specific prior (0 mean functions)." In our experiments we use a 0 mean prior for regression and a uniform prior for classification, **but the method can work with any arbitrary choice of the label prior.**
> >
> > ---
> >
> > 14. "Figure 3 is hard to read..." From the suggestion of another reviewer, **we limited the number of models in figure 3, and added the rest to the appendix.**
> >
> > We thank you for your review, and have incorporated changes as mentioned above.

---

> ### Author Response · Authors · 2020-11-23
> **Summary of response and changes**
>
> This is a summary of what we have provided in response to your feedback.
>
> - We have added a GP as a reference in figure 1.
> - We have included the idea of changing the amount of generated data based on the manifold geometry to section 6 as a
>   possible future work.
> - We have reconfigured figure 3 to only contain a subset of the models, and have added the rest of the models in the appendix
> - We have added an additional results studying the effect of PAD on different corruption intensities (figure 4)
>
> We have done our best to address every comment in the initial feedback. The end of the author discussion period is approaching, so if there are any further questions or concerns we would like to do our best to address them as soon as possible.
>
> Thank you for your feedback.

---

> ### Author Response · Authors · 2020-11-25
> **Added comparison to DUN**
>
> We have added a new experimental result comparing PAD to DUN [1]. The added tables are included in markdown below, and in comparison to all other models in the current version of the paper.
>
> ### Depth Uncertainty in Neural Networks [1]
>
> ### Negative Log Likelihood ($\downarrow$)
>
> | metric    | Housing | Concrete | Energy | Kin8nm | Naval | Power | Wine | Yacht |
> |-----------|---------|----------|--------|--------|-------|-------|------|-------|
> | DUN       | 4.86    | 4.78     | 4.41   | 0.85   | -1.16 | 4.34  | 3.61 | 6.37  |
> | DE PAD    | 3.61    | 5.13     | 3.24   | -0.38  | -2.81 | 3.12  | 1.24 | 3.78  |
> | R1BNN PAD | 3.84    | 4.30     | 3.78   | 0.09   | -3.25 | 3.13  | 1.27 | 4.14  |
> | SWAG PAD  | 3.80    | 4.47     | 3.51   | -0.58  | -2.46 | 3.18  | 1.20 | 3.53  |
> | MC PAD    | 4.32    | 4.98     | 3.34   | -0.53  | -0.96 | 3.18  | 1.22 | 3.45  |
>
> ### Expected Calibration Error ($\downarrow$)
>
> | metric    | Housing | Concrete | Energy | Kin8nm | Naval | Power | Wine | Yacht |
> |-----------|---------|----------|--------|--------|-------|-------|------|-------|
> | DUN       | 15.8    | 14.3     | 19.5   | 16.5   | 22.4  | 19.3  | 18.7 | 18.9  |
> | DE PAD    | 5.12    | 12.87    | 6.34   | 2.38   | 6.19  | 1.60  | 1.53 | 2.37  |
> | R1BNN PAD | 6.04    | 2.94     | 4.42   | 4.63   | 3.77  | 1.38  | 2.14 | 2.18  |
> | SWAG PAD  | 6.73    | 8.72     | 6.05   | 2.15   | 4.59  | 1.91  | 0.74 | 2.76  |
> | MC PAD    | 8.38    | 12.78    | 6.30   | 2.30   | 5.52  | 1.90  | 1.39 | 3.58  |
>
> We think it is important to note the following about this model:
>
> - Across all PAD augmented models, PAD outperforms DUN 37/40 times in terms of NLL and 40/40 times in terms of calibration error.
> - As DUN requires tuning the number of layers, we were unable to limit the number of layers to 2 as we did in the other experiments.
>
> Throughout the discussion period, we have added a number of extra experimental results which highlight the empirical effectiveness of PAD, and we have done our best to respond to each comment. If there are any remaining issues which are not clear, please do not hesitate to  ask us for clarification.
>
> Thank you for your time in reviewing our work,
>
> Authors
>
> [1] https://arxiv.org/abs/2006.08437

---

### Official Review · AnonReviewer2 · 2020-10-29
**Clever model-driven data augmentation solution for OOD predictions with promising results**

**Rating:** 6
**Confidence:** 3

**Review:**

I have read the authors' responses to all reviews and ultimately elected to leave my score as it is (weak accept). I think the empirical results are strong, and while I am not as troubled by the motivation and framing of the work as reviewers 3 and 4, I think their more conceptual and methodological critiques have merit, dampening my enthusiasm for the submission.

-----

This submission proposes a model-driven data augmentation strategy that aims to improve the calibration and reduce the over-confidence of a variety of Bayesian neural network architectures when dealing with out-of-distribution (OOD) samples. It involves adding a generator network that aims to generate plausible OOD samples during training, along with an objective term that tries to force the predictor network to make high entropy (low confidence) predictions for these samples. The paper does a fairly thorough empirical comparison with ten datasets (eight regression, two image classification) and half a dozen baselines, most of which can be combined with PAD. The results indicate that PAD usually improves both calibration and accuracy by at least a small amount.

This is a solid paper: the proposed method seems sensible (if pretty complex) and appears to be modestly effective in the included experimental results. The introduction summarizes the paper's contributions as:
1. It proposes a model-driven data augmentation technique aimed at improving calibration and reducing over-confidence for OOD samples.
2. It adapts and extends the technique to regression problems, which the paper argues is unprecedented.
3. It demonstrates empirically that the proposed approach improves the OOD accuracy and calibration of four different strong Bayesian neural net models.

I lack the broad familiarity with the data augmentation literature required to verify claim (2.). I suspect that if this simple claim is true, then it may be trivially so: it's hard to believe that _no one_ has applied data augmentation to regression tasks, so perhaps folks haven't bothered to publish it. The authors can always modify or remove this claim, if needed. The other two contributions seem supported, although the empirical improvements are for the most part small (and probably not statistically significant?). I lean weakly toward acceptance: I would not oppose its inclusion in the ICLR 2021 proceedings, but I wouldn't enthusiastically endorse it. I'll explain below.

The paper's motivation as laid out in Sections 1 and 2 is sound: calibration and proper quantification of uncertainty are increasingly important in a wide range of applications where machine learning has real world consequences for safety, fairness, etc. What is more, existing techniques based on neural networks (increasingly widespread) do seem to suffer significant flaws, especially exhibiting overconfidence when they should not. The paper offers a diagnosis in the form of a conjecture (Section 2.2): "failure to revert to the prior $p(\theta)$ for regions of the input space with insufficient evidence to warrant low entropy predictions." Figure 1 effectively visualizes this phenomenon in a toy setting, but no further proof is offered. Further, the assumption that prior reversion is the correct thing to do isn't examined (though that's a basic tenet Bayesian modeling, so we'll set that aside).

The proposed technique seems sensible, if complicated: add a generator network to produce OOD pseudo-samples during training and penalize the prediction network for making high confidence (low entropy) predictions on these pseudo-samples. We can consider this a form of model-driven (vs. heuristic) data augmentation. The generator loss, given in Equation (5), looks correct to my non-expert eye, and I suspect it's immediately comprehensible to readers familiar with GANs, VAEs, and Bayesian neural nets. The intuition for the OOD samples resonates with me: they should be close enough to real data to be plausible but far enough away that the predictor would be unjustified in assigning a high confidence or departing from the prior.

The regularization term in Equation (7) is a bit more arcane at first glance, but it's intuitive: the conditional prediction distribution should be close to the prior for pseudo-data points far from the real training distribution. The derivations of the K-L term for regression and categorical classification are given in the appendix, but these aren't critical details for judging the significance of the paper (they're quite straightforward).

The design of the experiments is sound: they simulate OOD settings by clustering each dataset and using distinct clusters for training and test splits and measure both accuracy and calibration. I don't have an opinion about the choice of the Kuleshov metric for calibration. The chosen baselines look strong, but I am not up-to-date on the relevant literature so I would not be able to identify a non-obvious missing baseline.

The experimental results are promising. A PAD variant is usually (but not always) the best for each task and metric (exceptions include GP for Naval/accuracy and R1BNN for Power/calibration). Perhaps more important PAD does generally seem to improve both accuracy and calibration across both variants (DE, MC, SWAG, etc.) and datasets. So in other words, if a modeler chooses to use one of the compatible Bayesian neural networks, in most cases they should also use PAD.

The work and manuscript have a few weaknesses that prevent me from more strongly recommending acceptance. For one, some of the exposition around training is unclear, in particular, how the objectivees in Equations (5) and (8) are combined during training.

I praised the results above, but I think the manuscript's interpretation of its results (Section 4.3) is still more generous than mine. PAD does consistently improve accuracy and calibration, but the margin is sometimes small, raising the question aboout whether the added complexity is worthwhile in all cases. The paper argues that PAD consistently improves the calibration curves in Figure 3, at least for poorly calibrated models, but that does not seem obvious to me. This might be because the curves are somewhat cluttered, but I see a number of exceptions where the PAD look potentially worse: Energy, Naval (SWAG), and Yacht.

I think perhaps the real problem is not the results themselves, which overall are strong, but rather the manuscript's rather cursory discussion of the results and its failure to offer any insights or guidance about, e.g., when PAD should be expected to help (based on task or dataset) or which baselines it works best with.

One last note: I don't want to over-index on a toy figure included for illustration purposes, but I don't find the results in Figure 1 convincing! Perhaps I am misunderstanding what behavior we desire (if so, please correct me). I agree that the PAD distribution does a better job of capturing the uncertainty in the central low-data area, but at the left- and righthand ends, the baselines actually look preferable in that the uncertainty is often appropriately wider and contains or at least follows the true function. It looks like PAD might be over-regularizing things in these cases.

Here are some actionable suggestions for improvements:
- Clarify how the objectives are combined and how training proceeds. Consider adding, e.g., an "Algorithm" summary figure.
- Expand the results discussion beyond simply restating the results (which are displayed in the Tables). For the raw accuracy and calibration numbers, perhaps you could compute some summary statistics for the baseline vs. PAD differences, so readers could get a quick sense of whether PAD usually beats the baseline. Maybe also some counts for how often a PAD variant has the best performance.
- Also in the discussion, try to distill out some illustrative patterns that could be turned into insights or practical guidance.
- For the calilbration curve plots (Figure 3), consider reducing the clutter by removing redundant curves. For most of the tasks, most baselines (and corresponding PAD variants) are quite similar so perhaps you could show a representative subset for each task (and then put the complete figures in the appendix).

Here's a laundry list of questions:
- How does training proceed? Is it a typical alternating adversarial optimization, i.e., optimize generator, then discriminator, repeat?
- What additional computational complexity does PAD introduce during training?
- Under which conditions PAD should be expected to help most: type or distribution of data, task structure, baseline model, etc.?

---

> ### Author Response · Authors · 2020-11-14
> **Expanded analysis, clarified figure 1**
>
> Thank you for taking the time to review our work. We will respond to each of your comments below in order.
>
> 1. "it's hard to believe that no one has applied data augmentation to regression tasks..." **we don’t state that data augmentation methods have never been used in regression, but only that our method of creating a distributional shift in the dataset for OOD testing purposes has not been proposed before** (much like that of [1], which proposes the ImageNet-C corrupted dataset for classification).
>
> ---
>
> 2. “Clarify how the objectives are combined and how training proceeds...” We give an algorithm in the appendix of the paper as well as the relevant code in the supplementary material which show that training proceeds in two steps where we first optimize the discriminative model with (8) which enforces that the discriminative model predicts the appropriate entropy, and then we go on to optimize the generative model with (5) in a separate gradient step. **In addition to the provided algorithm, we plan to release all code on Github pending acceptance.**
>
> ---
>
> 3. To address your concerns about what figure 1 actually shows, we think it is important to note a few things. 1) As you said, the central area of missing data has an appropriate increase in uncertainty in PAD models. 2) You mentioned that the baselines actually follow the true function with an increase in uncertainty. This is true in some cases, but the following of the true function is actually quite random as can be seen across the top row of figure one. Likewise, the behavior of the uncertainty in the top row is undefined outside of the training data, sometimes staying near constant and sometimes exploding. The explosion of uncertainty exhibited by some of the baselines is indeed preferable to constant or contracting uncertainty, which is why these models are strong baselines for OOD calibration, but this explosion is uninformative and inappropriate for Bayesian models which should revert towards the uncertainty of the prior for a random OOD datapoint. Furthermore, the explosion of uncertainty has a detrimental effect on the log likelihood of the models which don’t revert towards the prior, as can be seen in the $log p(y|x)$ numbers in figure 1. **To put it simply, beyond the boundary of the training data, the behavior of the baseline models is undefined while the behavior of PAD’d models is consistently predictable**. To illustrate our point about the unpredictability of the baselines in figure one, we re-ran the toy experiment, adding a GP as reference (figure 1). We included the original toy example (figure 8) in the appendix, and it can be seen that PAD models are mostly consistent between the two figures, while the baseline models are not.
>
> ---
>
> 4. “PAD does consistently improve accuracy and calibration, but the margin is sometimes small...”, We disagree with this statement and have **specifically underlined instances in table 1 where PAD decreases NLL by more than one on a log scale.** We point out reasons why this might only occur sometimes in the next point.
>
> ---
>
> 5. We would like to address the following three concerns together. “Under which conditions PAD should be expected to help most: type or distribution of data, task structure, baseline model, etc.?”, “...offer any insights or guidance about, e.g., when PAD should be expected to help (based on task or dataset)...”, and “Also in the discussion, try to distill out some illustrative patterns that could be turned into insights or practical guidance.” We think these can be addressed together by noting that it depends on how dense the data being predicted are and how likely it is that a test point may be OOD. This would be highly dependent on the specific dataset under consideration and would likely be subject to the opinion of domain experts. For example, **The Kin8nm and Power datasets are some of the lowest performers for PAD, and they also form a dense cluster as can be seen in figure 6 and 7 of the appendix. In such a dense cluster is unlikely that there is any relevant “dead space” between the data. We thank you for pointing this out, and we have added a relevant analysis and guidance to the discussion in section 4.3.**
>
> ---
>
> 6. "For the calilbration curve plots (Figure 3), consider reducing the clutter by removing redundant curves..." We have reduced the number of models in figure three and included the rest of the models in the appendix (figure 8).
>
> ---
>
> 7. **We have added more summary statistics to the analysis in section 4.3**
>
> ---
>
> 8. “What additional computational complexity does PAD introduce during training?...” **We have added this information to the end of section 3.**
>
> We thank you for your constructive feedback, we feel it has improved our submission.
>
> [1] Hendrycks, Dan, and Thomas Dietterich. "Benchmarking neural network robustness to common corruptions and perturbations." arXiv preprint arXiv:1903.12261 (2019).

---

> > ### Comment · AnonReviewer2 · 2020-11-23
> > **Nice revision**
> >
> > Authors,
> >
> > I want to commend your work during this response period: you advocated for your work but also accepted and acted upon constructive criticism, delivered a solid revision, and even ran new experimental results.
> >
> > Heading into the reviewer-only discussion, I will do my best to get to the bottom of some of the other reviewers' criticisms and to push them to separate out the substantive objections that might warrant rejection from nitpicks and opinion.
> >
> > Good luck!

---

> > > ### Author Response · Authors · 2020-11-24
> > > **Thank you for your continued evaluation**
> > >
> > > AnonReviewer2,
> > >
> > > Thank you for your further evaluation of our work. We have done our best to address every concern and critique in detail, and we appreciate your time and care in evaluating our work carefully. We will be available until the very end of the discussion period, so if there is anything which is not fully addressed in our response, please do not hesitate to ask us for clarification.
> > >
> > > Thank you,
> > >
> > > Authors

---

> ### Author Response · Authors · 2020-11-23
> **Summary of response and changes**
>
> This is a summary of what we have provided in response your feedback.
>
> - We have reconfigured figure 1 and provided a more in depth explanation of what is pictured
> - We have added extra analysis and summary statistics to section 4.3
> - We have reconfigured figure 3 by reducing the number of pictured models and adding the rest to the appendix
> - We have added an additional results studying the effect of PAD on different corruption intensities (figure 4)
>
> We have done our best to address every comment in the initial feedback. The end of the author discussion period is approaching, so if there are any further questions or concerns we would like to do our best to address them as soon as possible.
>
> Thank you for your feedback.

---

### Author Response · Authors · 2020-11-18
**Revision Update #1**

# Revision Update

- We have added a GP as reference in figure 1. We also re-ran the toy experiments in figure 1 and placed the original figure 1 in the appendix (figure 8), highlighting the unpredictability of OOD data on the baseline models.
- We have included an expanded analysis in section 4.3 to further evaluate where PAD is most effective
- We have added an additional experiment which evaluates PAD and the baselines on the uncorrupted test set as well as individually on 5 different corruption intensities for classification (figure 4, and figure 10)
- We have limited the number of models present in figure 3 and added the remainder of the models to figure 9 in the appendix.

We have updated other minor issues, and responded to each comment in the reviews. We are still working on a few remaining suggestions as noted in our individual responses. We thank the reviewers so far for taking time to review our work, and encourage further discussion if there are any further questions or concerns.

Thank you!

---

### Author Response · Authors · 2020-11-23
**The end of the discussion phase approaching**

Dear Reviewers,

Could you please go over our responses and the revision since we can have interactions with you only by this Tuesday (24th)? We have responded to your comments and faithfully reflected them in the revision, and provided additional experimental results that you have requested. We sincerely thank you for your time and efforts in reviewing our paper, and your insightful and constructive comments.

Thanks, Authors

---

### Author Response · Authors · 2020-11-23
**Revision Update #2**

# Revision Update

We have completed an additional real-world experiment which has been added to the appendix of the paper (section 8.3, tables 8 and 9). We used a dataset that consists of sales records for New York City spanning from 2008-2019, where we used the first two years as training/validation data and use the remaining years as test data (10 years). The large time frame evaluated creates a real temporal shift in the data, where it can be seen that PAD outperforms the baseline models in 35/40 instances in both negative log likelihood and calibration. We have attached the tables to *AnonReviewer1's* comment and included them in the most recent paper upload as well.

Thank you for the feedback! We are eager to answer any remaining comments from the reviewers.

---

> ### Comment · AnonReviewer2 · 2020-11-23
> **Nice work!**
>
> I plan to take a look shortly and will try to post follow-up questions ASAP so you have time to respond.

---

### Author Response · Authors · 2020-11-25
**Revision Update #3**

Dear Reviewers,

We have added a new experimental result comparing PAD to Depth Uncertainty in Neural Networks [1]. As we noted in our comment to *AnonReviewer4*, we think it is important to highlight that PAD augmented models outperformed DUN in 37/40 experiments in the table below in terms of NLL and 40/40 in terms of calibration error. For the full tables along with variance metrics, please refer to tables 1 and 2 in the pdf.

Throughout the discussion period, we have done our very best to address each and every concern raised by the reviewers. To summarize the most substantive changes, we have:

- Provided extra experimental results on the performance at varying degrees of OOD corruption intensity (figures 4 and 10)
- Provided an experiment on real-world OOD data cause by a temporal shift (tables 8 and 9)
- Added a new comparison baseline [1] (tables 1 and 2).
- Expanded the analysis in section 4.3
- Revamped figure 1, adding a GP, and clarifying how PAD is superior to the baseline models in the toy scenario.

A valid point from *AnonReviewers 3 and 4* is that the approach is an *ad-hoc solution*, which is defined by [2] as meaning *formed or used for specific or immediate problems or needs*. We agree that PAD arose as an intuitive solution to an outstanding problem which has thus far failed to be completely solved by more theoretic approaches in even the most recent BNN's. Through our experiments, **we have demonstrated that**:

- Recent BNN models still suffer from overconfidence.
- PAD delivers an effective solution to this problem which improves performance on OOD data in regression and classification datasets, as well as real-world scenarios.

To our knowledge such empirical works do fall into the category of papers that ICLR accepts and we believe that empirical studies contribute largely to ICLR’s success. We sincerely thank all the reviewers for their thoughtful comments, and for taking the time to carefully critique our work.

Thank you,

Authors
### Negative Log Likelihood ($\downarrow$)

| metric    | Housing | Concrete | Energy | Kin8nm | Naval | Power | Wine | Yacht |
|-----------|---------|----------|--------|--------|-------|-------|------|-------|
| DUN       | 4.86    | 4.78     | 4.41   | 0.85   | -1.16 | 4.34  | 3.61 | 6.37  |
| DE PAD    | 3.61    | 5.13     | 3.24   | -0.38  | -2.81 | 3.12  | 1.24 | 3.78  |
| R1BNN PAD | 3.84    | 4.30     | 3.78   | 0.09   | -3.25 | 3.13  | 1.27 | 4.14  |
| SWAG PAD  | 3.80    | 4.47     | 3.51   | -0.58  | -2.46 | 3.18  | 1.20 | 3.53  |
| MC PAD    | 4.32    | 4.98     | 3.34   | -0.53  | -0.96 | 3.18  | 1.22 | 3.45  |

### Expected Calibration Error ($\downarrow$)

| metric    | Housing | Concrete | Energy | Kin8nm | Naval | Power | Wine | Yacht |
|-----------|---------|----------|--------|--------|-------|-------|------|-------|
| DUN       | 15.8    | 14.3     | 19.5   | 16.5   | 22.4  | 19.3  | 18.7 | 18.9  |
| DE PAD    | 5.12    | 12.87    | 6.34   | 2.38   | 6.19  | 1.60  | 1.53 | 2.37  |
| R1BNN PAD | 6.04    | 2.94     | 4.42   | 4.63   | 3.77  | 1.38  | 2.14 | 2.18  |
| SWAG PAD  | 6.73    | 8.72     | 6.05   | 2.15   | 4.59  | 1.91  | 0.74 | 2.76  |
| MC PAD    | 8.38    | 12.78    | 6.30   | 2.30   | 5.52  | 1.90  | 1.39 | 3.58  |

[1] Antorán, J., Allingham, J., & Hernández-Lobato, J. M. (2020). Depth uncertainty in neural networks. Advances in Neural Information Processing Systems, 33.

[2] https://www.merriam-webster.com/dictionary/ad%20hoc

---

### Decision · Program_Chairs · 2021-01-07
**Final Decision**

**Decision:**

Reject

**Comment:**

This paper studies the problem of uncertainty estimation under distribution shift. The proposed approach (PAD) addresses this under-estimation issue, by augmenting the training data with inputs that the network has unjustified low uncertainty estimates, and asking the model to correct this under-estimation at those augmented datapoints. Results show promising improvement over a set of common benchmark tasks in uncertainty estimation, with comparisons to a number of existing approaches.

All the reviewer agreed that the experiments are well conducted and the empirical results are very promising. However, they also had a shared concern on the justification of the approach. Reviewers are less willing to accept a paper merely for commending its empirical performance.

I share the above concern as the reviewers, and I personally found the presentation of the approach a bit rush and disconnected from the motivation. For example, the current presentation feels like the method is motivated by BNNs but it is not clear to me how the proposed objective connects to the motivation. Also no derivation of the objective is included in either main text or appendix.

In revision, I would suggest a focus on improving the clarity and theoretical justification of the proposed objective function.